# AEAP: A Reinforcement Learning Actor Ensemble Algorithm with Adaptive Pruning

## Abstract

Actor ensemble reinforcement learning methods have shown promising performance on dense-reward continuous control tasks. However, they exhibit three primary limitations: (1) diversity collapse when using a shared replay buffer, often necessitating carefully tuned regularization terms; (2) computational overhead from maintaining multiple actors; and (3) computationally inefficient policy gradients when using stochastic policies in ensembles due to high-variance estimates, requiring approximations that may compromise performance. To address this third limitation, we restrict the ensemble to deterministic policies and propose Actor Ensemble with Adaptive Pruning (AEAP), a multi-actor deterministic policy gradient algorithm that tackles the remaining limitations through a two-stage approach. First, to alleviate diversity collapse, AEAP employs dual-randomized actor selection that decorrelates exploration and learning by randomly choosing different actors for both environment interaction and policy update. This approach also removes reliance on explicit regularization. Second, when convergence to homogeneous policies still occurs over time, computational efficiency is further achieved through adaptive dual-criterion pruning, which progressively removes underperforming or redundant actors based on critic-estimated value and action-space similarity. Although AEAP introduces four additional hyperparameters compared to TD3 (a baseline single-actor deterministic policy gradient algorithm), we provide two domain-agnostic parameter configurations that perform robustly across environments without requiring tuning. AEAP achieves superior or competitive asymptotic performance compared to baselines across six dense-reward MuJoCo tasks. On sparse-reward Fetch benchmarks, AEAP outperforms deterministic policy gradient methods but falls short of baseline stochastic policy gradient algorithms. When compared to fixed-size multi-actor baselines, AEAP reduces wall-clock time without sacrificing performance, establishing it as an efficient and reliable actor ensemble variant.

## 1 Introduction

Deep reinforcement learning (RL) has demonstrated significant potential across diverse continuous control domains, such as video games (Mnih et al., 2015; Silver et al., 2017), robotic manipulation (Clegg et al., 2018; Peng et al., 2018), and traffic optimization (Ault et al., 2020; Ault & Sharon, 2021). A large body of reinforcement learning algorithms (Fujimoto et al., 2018; Haarnoja et al., 2018) rely on a single agent to explore the environment via trial and error. However, single-actor exploration augmentation (Burda et al., 2018a; Ostrovski et al., 2017) often stalls in narrow regions of high-dimensional action spaces, leading to poor sample efficiency or even suboptimal convergence (Yang et al., 2022).

Actor ensemble (i.e., multiple actors or policies) methods have emerged as a promising direction for enhancing exploration by maintaining a diverse set of actors (Peng et al., 2020; Ren et al., 2021). Despite their empirical success, three critical limitations persist. First, shared replay buffers tend to accelerate diversity collapse, where diversity refers to the entropy in action distributions across actors (Fujimoto et al., 2018). To alleviate this, explicit regularization strategies such as entropy bonuses or divergence penalties can be employed but prove difficult to tune (Sheikh et al., 2022; Masood & Doshi-Velez, 2019). Insufficient regularization yields homogeneous policies whereas excessive regularization destabilizes learning (Zahavy et al., 2023).

Second, computational overhead scales linearly with ensemble size (Chen et al., 2021). Third, when treating stochastic ensemble policies as components of a probabilistic mixture, computing policy gradients becomes computationally inefficient due to high-variance gradient estimates that grow with the number of mixture components, requiring surrogate approximations that can compromise performance (Ren et al., 2021).

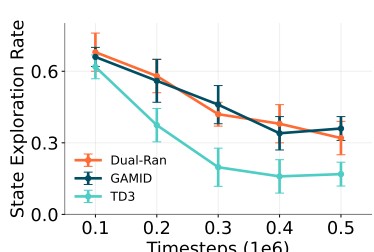

Motivated by these challenges and by recent evidence highlighting the benefits of deterministic policy gradients for training Gaussian-mixture actors (Dey & Sharon, 2024), we propose Actor Ensemble with Adaptive Pruning (AEAP), a multi-actor deterministic policy gradient algorithm. AEAP comprises two key components: (i) dual-randomized actor selection where one random actor interacts with the environment while another random actor receives gradient updates; and (ii) a dual-pruning mechanism based on both performance-estimated values and action-space distances.

Figure 1: Dual-randomized exhibits similar new state exploration rates as existing regularized method on Walker2d.

We demonstrate that dual-randomized actor selection maintains diversity by inducing high variance during exploration, without the need for explicit regularization. To validate this mechanism, we systematically compare four selection strategies in a 1-D bandit setting. After establishing the benefits in this simplified setting, we extend our analysis to realistic high-dimensional action spaces, and measure diversity through two metrics: (i) Principal Component Analysis (PCA) (Jolliffe, 1986) decomposition of action outputs during training; and (ii) new state exploration rate using RSNorm (Lee et al., 2025) to normalize and discretize states (i.e., tracking newly encountered discretized states over time). We compare against TD3 (Fujimoto et al., 2018) and a recent deterministic ensemble method Gaussian Mixture Deterministic Policy Gradient (GAMID) (Dey & Sharon, 2024), which includes explicit diversity regularization. As illustrated in Figure 1, dual-randomized selection maintains diversity at similar levels to existing regularized methods. (Further details are provided in Section 4.2).

Two observations motivate our following dual-pruning strategy: (i) high variance in updates inevitably produces unstable actors, and persistent low-performing actors would impede learning convergence, necessitating performance-based pruning, and (ii) computational waste stems from homogeneous policies or actor dominance, requiring redundancy-based pruning. We demonstrate that this pruning mechanism readily combines with existing ensemble methods (Figure 2.(a)) and independently improves performance.

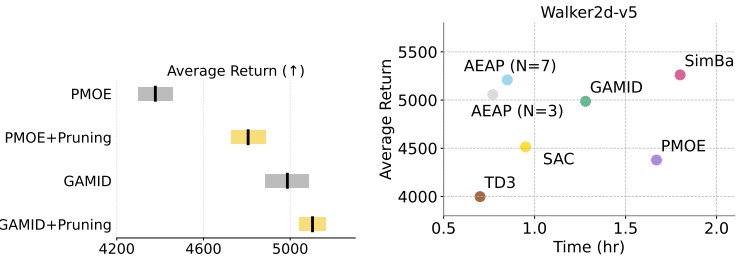

Figure 2: (a) Performance improvement in existing ensemble methods (GAMID/PMOE) after integrating pruning on Walker2d. (b) AEAP achieves high computational efficiency without sacrificing performance on Walker2d.

To evaluate the performance of the combination, we provide comprehensive empirical evaluations using two domain-agnostic parameter configurations: a conservative pruning approach for stable training, and an aggressive pruning approach for efficient training. Wall-clock time measurements confirm that AEAP achieves better computational efficiency than fixed-ensemble approaches without sacrificing performance (Figure 2.(b)). On dense-reward MuJoCo (Todorov et al., 2012) tasks, both settings achieve competitive performance that matches or exceeds baseline algorithms. On sparse-reward Fetch tasks, both outperform deterministic policy gradient baselines but underperform stochastic policy gradient methods Soft Actor-Critic (SAC) (Haarnoja et al., 2018) and SimBa (Lee et al., 2025), with aggressive pruning demonstrating superior performance relative to conservative pruning. By integrating dual-randomized actor selection with adaptive dual-criterion pruning, our work addresses the tension between exploration benefits and computational efficiency, offering a practical, efficient, and reliable alternative to fixed-size actor ensembles.

## 2 Related Work

We review the most relevant actor ensemble methods herein, and relegate a detailed discussion of single-actor exploration and critic ensembles to Appendix A. Actor ensemble methods can enhance exploration by maintaining multiple policies, but face fundamental trade-offs between actor diversity and computational efficiency. We categorize these approaches by their coordination mechanisms.

**Distributed RL.** A natural extension to enhance exploration is employing multiple actors concurrently. Distributed RL frameworks maintain multiple homogeneous actors generating parallel trajectories, with a central learner aggregating experiences for policy updates (Mnih et al., 2016; Espeholt et al., 2018; Horgan et al., 2018). Kapturowski et al. (2019) augment this paradigm with recurrent networks and burn-in replay, while Espeholt et al. (2020) centralise inference on GPUs for higher throughput. However, these approaches demand substantial computational resources and introduce significant synchronization overhead between learners and actors, they require access to an emulator, which is often impractical in real-world settings.

**Shared Replay Buffer.** A different approach maintains a *shared* replay buffer across actors, assuming all actors contribute throughout training. These methods diverge in how they maintain actor diversity. Value-based selection approaches guide actor specialization with heuristics. Previous work partitions the state space and train specialized actors on subtasks (Ghosh et al., 2018) and aggregates outputs from multiple policies to stabilize training (Barth-Maron et al., 2018; Chen & Peng, 2019; Januszewski et al., 2021; Li et al., 2023). Zhang et al. (2018b) combine actor ensembles with tree search for refined action selection. Yet these methods still suffer from diversity collapse as actors often converge when sharing experiences.

Explicit regularization methods add diversity-preserving terms to the objective. These include behavioral penalties (Peng et al., 2020; Zahavy et al., 2023), pairwise KL divergence maximization (Sheikh et al., 2022), and mixture entropy optimization (Baram et al., 2021; Ren et al., 2021; Peng et al., 2019). Gamid (Dey & Sharon, 2024) extend this line by applying deterministic policy gradients to Gaussian mixtures, achieving strong performance in continuous control tasks. However, these approaches remain sensitive to hyperparameter tuning and can still exhibit single-actor dominance despite explicit diversity mechanisms. Our work extends Gamid by introducing dual-randomized actor selection and adaptive pruning to eliminate redundant actors while preserving exploration benefits.

**Population-based RL.** A mutation-based approach generates new actors by mutating high-performing ones, inherently promoting diversity. This approach is exemplified by several works (Conti et al., 2018; Doncieux & Mouret, 2013; Lehman & Stanley, 2011), but requires careful tuning of evolutionary parameters and often suffers from sample inefficiency in continuous control tasks.

**Pruning in RL.** Works on pruning in RL primarily target network compression to enhance computational efficiency and improve generalization (Song et al., 2019; Zhang et al., 2018a). Studies demonstrate that magnitude-based and structured pruning can significantly compress deep RL networks while preserving or enhancing performance (Livne & Cohen, 2020; Graesser et al., 2022; Obando-Ceron et al., 2024; Park et al., 2024). Our method differs by disabling entire actor networks, addressing ensemble-level redundancy rather than within-network sparsity.

*Remark* 1. We clarify that target network approaches (van Hasselt et al., 2015; Fujimoto et al., 2018), while related, fundamentally differ from ensemble-based techniques. Target networks stabilize training by decoupling rapidly changing networks from their bootstrapped targets, thus making updates more predictable and reducing oscillations. In contrast, ensemble-based approaches explicitly promote diversity or exploration through maintaining multiple distinct policies or critics.

## 3 Preliminaries

**Deep Reinforcement Learning.** The goal of reinforcement learning is to find a policy that maximizes the expected cumulative discounted return over a long horizon for a given Markov Decision Process (MDP) defined by the tuple $(\mathcal{S}, \mathcal{A}, \mathcal{P}, R, \gamma)$, where $\mathcal{S}$ is the state space, $\mathcal{A}$ is the action space. The transition dynamics are captured by $\mathcal{P} : \mathcal{S} \times \mathcal{A} \to \Delta(\mathcal{S})$, where $\Delta(\mathcal{S})$ denotes the space of probability distributions over $\mathcal{S}$. The reward function is given by $R : \mathcal{S} \times \mathcal{A} \to \mathbb{R}$, and $\gamma \in [0, 1)$ is the discount factor (Sutton & Barto, 1998).

A policy can be either stochastic $\pi : \mathcal{S} \to \Delta(\mathcal{A})$, mapping states to distributions over actions, or deterministic $\mu : \mathcal{S} \to \mathcal{A}$, mapping states directly to actions. The action-value function under policy $\pi$ is defined as: $Q^\pi(s, a) = \mathbb{E}_{\tau \sim \pi} \left[ \sum_{t=0}^{\infty} \gamma^t R(s_t, a_t) \,\middle|\, s_0 = s, a_0 = a \right]$ where the expectation is over trajectories $\tau$ generated by following policy $\pi$. The optimal action-value function, $Q^*(s, a) = \max_\pi Q^\pi(s, a)$, satisfies the Bellman optimality equation (Bellman, 1957): $Q^*(s, a) = R(s, a) + \gamma \mathbb{E}_{s' \sim P(\cdot|s,a)} [\max_{a' \in \mathcal{A}} Q^*(s', a')]$. In a vast body of value-based deep reinforcement learning, the Q-function is often approximated using neural networks with parameters $\theta$, denoted as $Q_\theta$ (Mnih et al., 2015). The network is trained by minimizing the temporal difference error over transitions $(s, a, r, s')$ sampled from a replay buffer $\mathcal{D}$ (Sutton, 1988): $L(\theta) = \mathbb{E}_{(s,a,r,s') \sim \mathcal{D}} \left[ (r + \gamma \max_{a' \in \mathcal{A}} Q_{\theta'}(s', a') - Q_\theta(s, a))^2 \right]$ where $\theta'$ denotes target network parameters that are periodically updated to stabilize training.

**Deterministic Policy Gradient.** The Deterministic Policy Gradient (DPG) theorem (Silver et al., 2014) provides a framework for optimizing deterministic policies $\mu_\phi : \mathcal{S} \to \mathcal{A}$. For a parameterized deterministic policy $\mu_\phi$ with parameters $\phi$, the objective is to maximize the expected return: $J(\phi) = \mathbb{E}_{s \sim \rho^\mu}[R_0] = \mathbb{E}_{s \sim \rho^\mu}[Q^\mu(s, \mu_\phi(s))]$ where $\rho^\mu$ is the state distribution under policy $\mu$. The deterministic policy gradient is given by: $\nabla_\phi J(\phi) = \mathbb{E}_{s \sim \rho^\mu} \left[ \nabla_a Q^\mu(s, a)|_{a=\mu_\phi(s)} \nabla_\phi \mu_\phi(s) \right]$.

Deep Deterministic Policy Gradient (DDPG) (Lillicrap et al., 2019) implements DPG using deep neural networks for approximating both the policy $\mu_\phi$ and critic $Q_\theta$. The critic is trained by minimizing the temporal difference error: $L(\theta) = \mathbb{E}_{(s,a,r,s') \sim \mathcal{D}} \left[ (y - Q_\theta(s, a))^2 \right], \quad y = r + \gamma Q_{\theta'}(s', \mu_{\phi'}(s'))$. Twin Delayed Deep Deterministic Policy Gradient (TD3) (Fujimoto et al., 2018) addresses overestimation bias in DDPG through three modifications: (1) clipped Double Q-learning, using twin critics $Q_{\theta_1}, Q_{\theta_2}$ where the minimum is used for value estimation, (2) delayed policy updates where the actor is updated less frequently than critics (typically every $d$ iterations), and (3) target policy smoothing, adds clipped noise to target actions to regularize the value function:

$$y = r + \gamma \min_{i=1,2} Q_{\theta_i'}(s', \mu_{\phi'}(s') + \varepsilon), \quad \varepsilon \sim \text{clip}(\mathcal{N}(0, \sigma), -c, c)$$

$$\theta' \leftarrow \alpha\theta + (1 - \alpha)\theta', \quad \phi' \leftarrow \alpha\phi + (1 - \alpha)\phi'$$

where $\alpha$ is the soft update rate. These modifications often lead to more stable learning and improved performance in continuous control tasks.

**Gaussian Mixture Policies.** While standard policy gradient methods optimize a single Gaussian policy $\pi(a|s) = \mathcal{N}(a; \mu(s), \Sigma)$. Gaussian mixture models (GMMs) (Alspach & Sorenson, 1972) offer richer representational capacity through multiple components: $\pi(a|s) = \sum_{i=1}^{N} w_i(s) \mathcal{N}(a; \mu_i(s), \Sigma_i)$ where $w_i(s)$ are state-dependent mixing weights satisfying $\sum_i w_i(s) = 1$. However, computing exact policy gradients for stochastic mixtures requires intractable marginalization over component assignments. Gamid (Dey & Sharon, 2024) circumvents this by treating each component mean $\mu_i$ as a deterministic policy, enabling direct application of deterministic policy gradients to GMMs.

## 4 Algorithm

In this section, we present Actor Ensemble with Adaptive Pruning (AEAP) and analyze its two key components: dual-randomized actor selection and adaptive dual-criterion pruning.

### 4.1 AEAP

AEAP maintains a dynamic ensemble of active actors that evolves throughout training, as shown in Algorithm 1. At each iteration, an actor is randomly selected from the active ensembles, perturbing its actions with the Gaussian noise for environment interaction, and collecting transitions for the shared replay buffer. For policy update, a different actor is randomly selected from the active ensembles and updated via deterministic policy gradients. Every $\kappa$ iterations, the pruning mechanism evaluates all actors against our dual criteria, eliminating those that are either underperforming (low Q-values) or redundant (similar behaviors).

---

**Algorithm 1** AEAP: **A**ctor **E**nsemble with **A**daptive Actor **P**runing

---

1: **AEAP Hyperparameters:** (1) Initial number of actors $N$; (2) Distance pruning threshold $\zeta$;
2:                            (3) Critic pruning ratio threshold $\xi$; (4) Pruning frequency $\kappa$
3: **TD3 Hyperparameters:** Exploration variance $\Sigma$; Target update rate $\tau$; Policy update frequency $d$
4: **Initializations:** Replay buffer $\mathcal{D}$; Actor networks $\{\phi_i\}_{i=1}^N$;
5:                Critic networks $\theta_1, \theta_2$ and targets $\theta'_1, \theta'_2$; Active actor set $\mathcal{S}_{\text{active}} \leftarrow \{1, \ldots, N\}$
6: **for** $t = 1$ to $T$ **do** $\Rightarrow$ TD3-style Training with Stochastic Actor Selection
7:     Select actor $i \sim \text{Uniform}(\mathcal{S}_{\text{active}})$ and sample action $a \leftarrow \pi_{\phi_i}(s) + \varepsilon, \varepsilon \sim \mathcal{N}(\mathbf{0}, \Sigma)$
8:     Execute $a$, observe $s', r, \text{done}$, and store $(s, a, r, s', \text{done})$ in $\mathcal{D}$
9:     Sample mini-batch $\mathcal{B}$ from $\mathcal{D}$
10:     Compute targets: $j \sim \text{Uniform}(\mathcal{S}_{\text{active}}), a' \leftarrow \pi_{\phi_j}(s') + \text{clip}(\varepsilon)$   , $y \leftarrow r + \gamma(1 - \text{done}) \min_{i=1,2} Q_{\theta'_i}(s', a')$
11:     Update all critics: $\theta_i \leftarrow \arg\min_{\theta_i} \frac{1}{|\mathcal{B}|} \sum_{\mathcal{B}} (Q_{\theta_i}(s, a) - y)^2$
12:     **if** $t \% d == 0$ **then**
13:         Update actor *randomly*: $\phi_k \leftarrow \arg\max_{\phi_k} \frac{1}{|\mathcal{B}|} \sum_{s \in \mathcal{B}} Q_{\theta_1}(s, \pi_{\phi_k}(s))$   $k \sim \text{Uniform}(\mathcal{S}_{\text{active}})$
14:         Soft-update critic targets: $\theta'_i \leftarrow \tau\theta_i + (1 - \tau)\theta'_i$
15:     **end if**
16:     **if** $t \% \kappa == 0$ **and** $|\mathcal{S}_{\text{active}}| > 1$ **then** $\Rightarrow$ AEAP Actor Pruning Mechanism
17:         Compute pairwise distances: $D_{ij} = \frac{1}{|\mathcal{B}|} \sum_{s \in \mathcal{B}} \|\pi_{\phi_i}(s) - \pi_{\phi_j}(s)\|_2$ for all $i \neq j$
18:         Compute actor values: $Q_i = \frac{1}{|\mathcal{B}|} \sum_{s \in \mathcal{B}} \min Q_{\text{targ}}(s, \pi_{\phi_i}(s))$ for all $i \in S_{active}$
19:         **if** $\exists z \in \mathcal{S}_{active}$ such that $Q_z < \xi \cdot \max_i Q_i$ **then** $\Rightarrow$ Performance pruning
20:             Remove actor with the lowest $Q$-value
21:         **else if** $\max_{i,j \in \mathcal{S}_{active}, i \neq j} D_{ij} < \zeta$ **then** $\Rightarrow$ Redundancy pruning
22:             Remove actor with lower $Q$-value from the closest pair
23:         **end if**
24:     **end if**
25: **end for**

---

## 4.2 Exploration Benefits of Dual-Randomized Actor Selection

Empirical studies (Ren et al., 2021; Dey & Sharon, 2024) and theoretical analyses (Calinon et al., 2012; Chan et al., 2022) demonstrate that actor ensembles enhance exploration, particularly where the state-action space remains largely unexplored by single-actor methods. Theoretical analyses examine exploration gains through entropy measures and probability ratios, highlighting that certain regions of the action space have substantially lower probabilities of being sampled by a single Gaussian actor than by an ensemble modeled as a Gaussian mixture. We further substantiate these insights through intuitive visual analyses, centered on two primary observations:

**Single-actor Exploration is Limited.** Merely increasing the exploration variance ($\sigma$) of a single actor, as commonly practiced in TD3, is insufficient to guarantee robust exploration, particularly in high-dimensional action spaces. We examine this exploration limitations of TD3 using the Humanoid environment from MuJoCo (Todorov et al., 2012), which features a challenging 17-dimensional action space. Figure 3 shows Principal Component Analysis (PCA) (Jolliffe, 1986) of actions generated by TD3 during the initial 50,000 training steps with varying exploration variance $\sigma \in \{0.1, 0.5\}$ and the distribution heatmaps for the first two action dimensions. The explained variance ratios reveal that TD3 concentrates exploration predominantly along a few dimensions. While raising $\sigma$ from 0.1 to 0.5 partially addresses this limitation, the concentration effect persists, presumably due to the action clipping that constrains noise augmented actions to valid bounds. Appendix B.2 provides additional

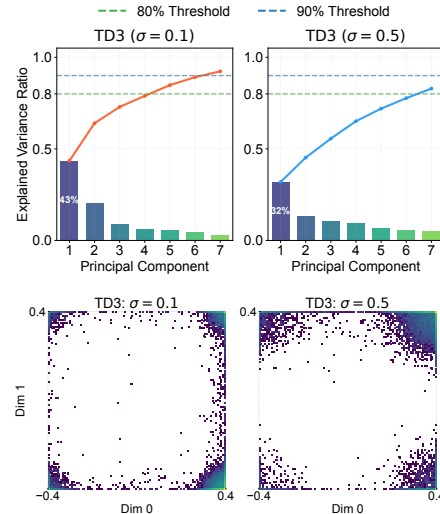

Figure 3: (a) PCA of TD3 actions with varying exploration variance on Humanoid. (b) TD3 action distribution heatmaps on Humanoid(first two dimensions).

**Actor Positions & Gradient Effects ( Step# 1000 )**

**Actor Positions & Gradient Effects ( Step# 5000 )**

**Actor Positions & Gradient Effects ( Step# 10000 )**

● Dual-Random ● Random+All ● Greedy+Random ● Greedy+All

**Action Space Visitation (3K Steps)**

| | -1 | -0.5 | | 0 | | 0.5 | | 1.0 |
|---|---|---|---|---|---|---|---|---|
| Dual-Random | 317 | 515 | 363 | 403 | 424 | 501 | 316 | 161 |
| Random+All | 157 | 166 | 227 | 500 | 684 | 881 | 189 | 196 |
| Greedy+Random | 172 | 214 | 313 | 471 | 562 | 706 | 381 | 181 |
| Greedy+All | 178 | 157 | 193 | 462 | 770 | 852 | 235 | 153 |

Figure 4: (a) The background gray curve shows the bimodal reward landscape in 1-dimensional action space. Colored dots represent actor positions at different training steps, with arrows indicating gradient-directed movement (From $-1$ to $1$). Methods with random updates (Dual-Random, Greedy+Random) update only one actor per step, with dual-randomized selection exhibiting multimodal behavior. (b) Cumulative action space visitation frequency over 3K steps.

results across multiple seeds, even with high exploration noise ($\sigma = 1.0$), TD3 exhibits clustering near the action space boundaries, limiting effective exploration in high-dimensional action spaces.

*Remark* 2. We provide additional similar analysis for the Hopper domain, a simpler environment with a 3-D action space, in Appendix B.1. These results indicate that increasing exploration noise in lower-dimensional settings can indeed broaden action coverage more effectively, highlighting the importance of dimensionality considerations when designing exploration strategies.

**Dual-randomized Actor Selection Maintains Diversity.** Inspired by dropout techniques in Hiraoka et al. (2022) that randomly deactivate $Q$-heads to prevent single critic dominance, we extend this randomization concept to actor selection: we randomly choose one actor interacts with the environment and another random one receives gradient updates. We evaluate four configurations by crossing two design choices: (1) how actors are selected for environment interaction — either randomly or through performance-based greedy selection, and (2) which actors receive policy updates — either a single randomly selected actor or all actors simultaneously. This yields four variants allowing us to isolate the impact of randomization at each stage, demonstrating that dual-randomized actor selection offers superior exploration advantages.

Using the continuous bandit problem from Huang et al. (2023), which features a deterministic reward function with two modes in a 1-D action space. We employ four actors with identical network architectures and orthogonal initializations. (Full experiment setting is reported in Appendix B.3).

Figure 4.(a) tracks current actor positions and corresponding gradient dynamics over time, revealing that configurations where all actors receive updates strongly converge toward a single local optimum. Conversely, dual-randomized selection exhibits multi-modal behavior capable of overcoming local optima.

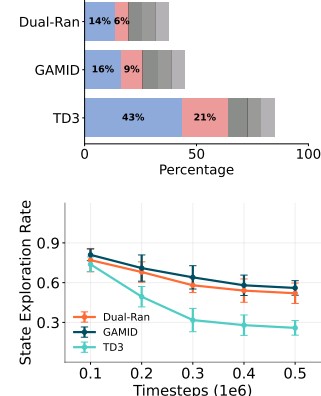

We attribute this to two factors: (i) random actor selection promotes diverse environment interactions, particularly early in training when the critic remains immature, thereby providing richer for critic learning, similar to supervised learning where balanced samples accelerate learning; and (ii) updating a single random actor introduces delayed, high-variance learning dynamics, giving actors that discover narrow high-reward regions a high probability of remaining there and avoiding premature convergence. As illustrated in Figure 4.(b), dual-randomized selection achieves the broadest action space coverage among all strategies. Cumulative action selection probabilities in Appendix B.3 confirm consistent patterns.

Figure 5: (b) PCA decomposition on Humanoid. (b) New state exploration on Humanoid.

This capability can also extends to high-dimensional action spaces. We demonstrate this using GAMID (Dey & Sharon, 2024), a multi-actor algorithm with explicit divergence regularization. Figure 5.(a) compares PCA decomposition of TD3, dual-randomized selection, and GAMID on Humanoid, showing the first five principal components where dual-randomized exhibits the most uniform distribution across dimensions. Figure 1.(b) demonstrates that diversified actions increase new state discovery, measured using Running Statistics Normalization (RSNorm) from (Lee et al., 2025) to standardize and discretize observations for

Table 1: Actor selection ratios and final mean performance($R$) for Gamid under varying $\varepsilon$ values. Actor labels are assigned by selection frequency, with Actor 0 being the most frequently selected.

(a) WALKER2D-V5

| $\varepsilon$ | Actor 0 | Actor 1 | Actor 2 | R |
|---|---|---|---|---|
| $1/N$ | 72.22% | 15.82% | 11.96% | 4,213 |
| $1/(2N)$ | 84.23% | 7.15% | 8.61% | 4,337 |
| $1/(10N)$ | 96.47% | 2.23% | 1.29% | 4,581 |

(b) HUMANOID-V5

| $\varepsilon$ | Actor 0 | Actor 1 | Actor 2 | Actor 3 | R |
|---|---|---|---|---|---|
| $1/N$ | 64.18% | 16.94% | 11.33% | 7.55% | 4,938 |
| $1/(2N)$ | 78.26% | 9.81% | 7.42% | 4.51% | 5,016 |
| $1/(10N)$ | 93.84% | 2.94% | 1.96% | 1.26% | 5,219 |

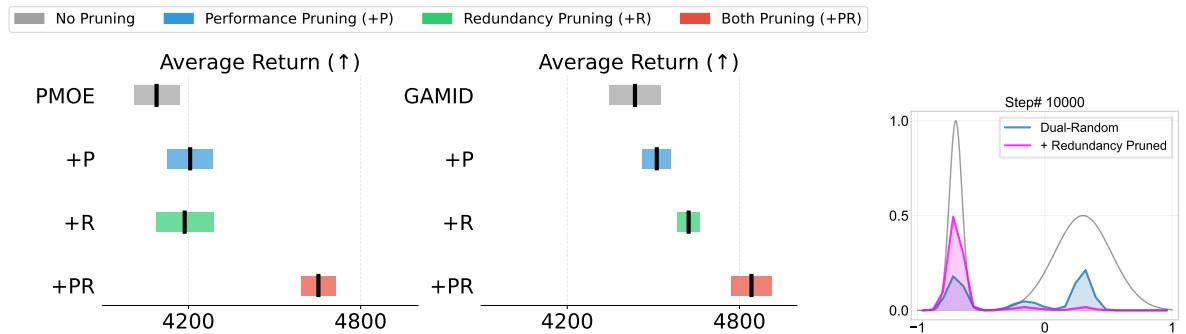

Figure 6: (a) Integration of pruning strategies leads to performance improvements for both GAMID and PMOE in Ant, with the highest gains achieved when both redundancy-based and performance-based criteria are combined. (b) Comparison of cumulative action selection probabilities between dual-randomized actor selection and its performance-based pruning extension.

tracking exploration rates (details in Appendix B.4). Dual-randomized exhibits similar new state exploration rates as existing regularized method on Humanoid.

However, this high variance has a double-edged effect: while enabling escape from local optima, unstable gradients may trap some actors in suboptimal regions between peaks, as shown in Figure 4.(b). This high visitation frequency between peaks arises because immature critic functions could generate scattered gradients, trapping actors in intermediate areas. We address this challenge in the following section.

### 4.3 Actor Pruning: From Exploration to Exploitation

We address two central questions regarding actor pruning strategy: (1) why pruning actors is preferable to retaining all actors throughout training, and (2) how actors should be pruned effectively.

**Pruning is preferable.** Two computational inefficiencies motivate pruning. First, as shown in Section 4.2, unstable gradients produce underperforming actors that impede learning convergence. Second, even with explicit regularization, training dynamics drift toward actor dominance where certain policies dominate sampling, wasting computation on seldom-used actors.

We demonstrate the second issue using GAMID, which selects actors for environment interaction via an $\varepsilon$-greedy rule. Table 1 reports the empirical selection ratios and final mean performance for each actor under three $\varepsilon$ values in Walker2d-v5 and Humanoid-v5. Smaller $\varepsilon$ yields better returns but also steeper concentration of interaction counts on one actor. A plausible explanation is that, maintaining several actors benefits exploration in the early phase when state–action visitation is sparse. However, as exploration becomes less valuable relative to exploitation, higher $\varepsilon$ enforces more random selections when the algorithm should exploit the superior actor, and thus leads to worse performance. This observation confirms that the computational overhead of maintaining all actors becomes increasingly unjustified, as remaining actors contribute little beyond redundant updates, motivating our adaptive pruning approach. Figure 6.(a) shows performance improvement in GAMID after integrating redundancy-based pruning.

**Pruning Strategy.** We introduce dual pruning based on performance-estimated values and action-space distances to maintain exploration diversity while ensuring computational efficiency:

- Performance-Based Pruning: Actors whose mean $Q$-value falls below a predefined ratio of the best performer are removed.

- Redundancy-Based Pruning: When maximum pairwise action distance falls below a threshold, the actor with lower $Q$-value is removed.

We detail hyperparameter choices for pruning and their rationale. The performance threshold $\xi = 0.85$ eliminates actors performing below 85% of the best performer, a value that empirically performs well across environments while avoiding overly aggressive pruning. A potential concern is that overestimation bias might incorrectly prune actors with better Q-value estimates. We address this in Appendix B.5 and find that several mitigating factors make this situation rare in practice. To ensure adequate training before pruning decisions, we use a pruning frequency of $\kappa = 10,000$ steps. The distance pruning threshold is defined as: $\zeta = c \cdot \sqrt{\dim(\mathcal{A})} \cdot a_{\max}$, where $\mathcal{A}$ is the action space with a maximum action magnitude $a_{\max}$ per dimension, and $c$ is a scaling constant. This directly reflects the maximal possible action-space distance.

We identify two particularly effective configurations:

1. **Conservative setting:** $(N = 3, c = 1.0)$ Small initial ensemble coupled with a conservative threshold ($\zeta = 1.0 \cdot \sqrt{\dim(\mathcal{A})} \cdot a_{\max}$), typically retains one or two trained actors for stability.

2. **Aggressive setting:** $(N = 7, c = 1.5)$ Large initial ensemble alongside a higher threshold ($\zeta = 1.5 \cdot \sqrt{\dim(\mathcal{A})} \cdot a_{\max}$), rapidly pruning to single dominant policy after leveraging exploration capabilities. This is particularly beneficial in sparse-reward environments.

We demonstrate the effectiveness of performance-based pruning using the previous bandit example. Figure 6.(b) shows that starting from three actors, with a performance ratio threshold of 0.85 and pruning frequency of 1,000 steps. Actors are progressively removed when their average $Q$-value falls below 85% of the best performer. This adaptive mechanism successfully guides the ensemble to the global optimum by removing suboptimal actors while preserving the best.

Figure 6.(a) demonstrates that integrating either single or combined pruning mechanisms into GAMID and another ensemble method Probabilistic Mixture-of-Experts SAC (PMOE)(Ren et al., 2021), yields performance in the Ant environment, which has a higher-dimensional action space. Additional comparisons in Humanoid and HalfCheetah are provided in Appendix B.6. Further ablation studies on AEAP, presented in Section 5.2, also analyze the two pruning strategies in detail and confirm that both contribute to performance improvements.

*Remark* 3. We briefly provide a heuristic justification for our preference of the maximum-distance criterion over a mean-distance criterion. Mean pairwise distances are known to concentrate in high-dimensional spaces, making them numerically indistinct and insensitive to meaningful behavioral differences, especially in cases involving small variance (Beyer et al., 1999). Reinforcement learning typically employs small variance in exploration, where subtle yet significant differences in actor behaviors are often undetectable by mean-based metrics. Consequently, we adopt a criterion based purely on the geometric properties of the action space, independent of variance, to more reliably identify genuine behavioral redundancy among actors.

## 5 Experiments

We aim to evaluate four aspects of AEAP: (1) performance competitiveness against established single-actor and multi-actor baselines across dense and sparse reward environments, (2) computational efficiency gains through adaptive pruning compared to fixed-size ensembles, (3) robustness of our proposed hyperparameter configurations across diverse continuous control tasks, and (4) effectiveness of individual and combined pruning strategies.

### 5.1 Baseline Comparisons and Computational Efficiency

**Baselines.** We evaluate AEAP against established baselines on continuous control benchmarks from MuJoCo (Todorov et al., 2012) and Fetch (Plappert et al., 2018a) domains. Our comparisons include

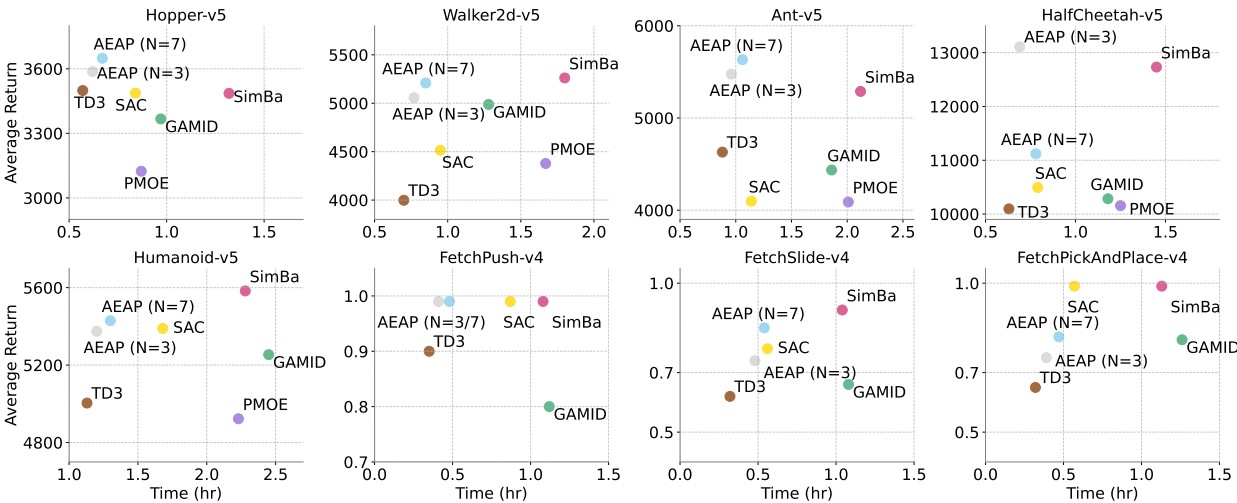

Figure 7: Performance-efficiency trade-off on MuJoCo continuous-control benchmarks (average episode return) and sparse-reward Fetch domain (success rate). While AEAP does not achieve the highest performance on every task, it consistently attains competitive performance across domains while requiring less wall-clock time than the recent state-of-the-art SimBa and other fixed-size actor ensembles, making it an efficient alternative when resources or training time are constrained. Wall-clock measurements represent mean runtime over seven trials (1M steps each) on NVIDIA RTX 4090 GPU and Intel i9-13900K CPU.

single-actor methods TD3 (Fujimoto et al., 2018) and SAC (Haarnoja et al., 2018), for multi-actor ensembles, we use Probabilistic Mixture-of-Experts SAC (PMOE) (Ren et al., 2021) and Gaussian Mixture Deterministic Policy Gradient (GAMID-PG) (Dey & Sharon, 2024). For completeness, we also include the most recent state-of-the-art deep RL architecture, SimBa (Lee et al., 2025), and observe that we achieve comparable post-training performance on over half of the MuJoCo domains with lower running time.

**Implementations.** We adopt the aggressive setting mentioned in Section 4.1 by default, implementations for TD3 and SAC utilize Stable-baselines3 (Raffin et al., 2021), whereas PMOE, GAMID-PG and SimBa leverage author-provided code, adhering strictly to recommended hyperparameters from original literature (detailed in Appendix C). Consistent with prior work (Ibarz et al., 2021), methods were augmented with Hindsight Experience Replay (HER) (Andrychowicz et al., 2018) for sparse reward tasks within the Fetch domains. All experiments are conducted over 7 random trials.

**Performance.** The post-training performance are summarized in Table 2, with corresponding learning curves illustrated in Appendix B.7. Figure 7 presents the results, where the x-axis represents computation time and the y-axis denotes performance. Points in the upper-left corner indicate higher performance and compute efficiency. On dense-reward Mujoco tasks, AEAP consistently attains competitive performance across domains while requiring less wall-clock time than recent state-of-the-art methods such as SimBa and other fixed-size actor ensembles or simpler baselines like SAC, even though it does not achieve the highest performance on every individual task. When resources or training time are constrained, AEAP offers a compelling performance-efficiency trade-off for practitioners prioritizing faster experimentation or deployment cycles. On sparse-reward Fetch tasks, AEAP substantially outperforms other deterministic methods while approaching the performance of SAC and SAC-based SimBa. The underperformance on FetchPickAndPlace likely stems from its multi-stage exploration challenge: SAC with entropy regularization keeps sampling diverse actions long enough to discover that sequence, whereas the deterministic actors in AEAP narrow their exploration once pruning begins, reducing the chance of stumbling onto the critical manipulation pattern.

**Efficiency.** Figure 7 demonstrates AEAP's efficiency gains through adaptive pruning. Despite starting with 7 actors, AEAP achieves substantially shorter runtime compared to fixed-size actor ensembles such as Gamid with competitive performance. This demonstrates the effectiveness of AEAP as a computationally efficient and reliable alternative to conventional multi-actor reinforcement learning algorithms.

Table 2: Mean performance and the 1-standard deviation on continuous control benchmarks. The best and second-best performing RL algorithms have been highlighted.

| | TD3 | SAC | GAMID | PMOE | SimBa | AEAP(N=3) | AEAP(N=7) |
|---|---|---|---|---|---|---|---|
| *MuJoCo (v5)* | | | | | | | |
| HalfCheetah | $10,097 \pm 476$ | $10,493 \pm 79$ | $10,285 \pm 253$ | $10,156 \pm 312$ | $12,732 \pm 345$ | $\mathbf{13,110} \pm 285$ | $11,121 \pm 429$ |
| Hopper | $3,499 \pm 225$ | $3,487 \pm 11$ | $3,367 \pm 196$ | $3,124 \pm 189$ | $3,486 \pm 370$ | $3,587 \pm 365$ | $\mathbf{3,649} \pm 358$ |
| Walker2d | $3,998 \pm 236$ | $4,514 \pm 12$ | $4,987 \pm 348$ | $4,378 \pm 254$ | $\mathbf{5,261} \pm 485$ | $5,057 \pm 275$ | $5,209 \pm 467$ |
| Ant | $4,630 \pm 475$ | $4,098 \pm 213$ | $4,436 \pm 292$ | $4,089 \pm 387$ | $5,288 \pm 195$ | $5,476 \pm 205$ | $\mathbf{5,632} \pm 199$ |
| Humanoid | $5,004 \pm 429$ | $5,389 \pm 23$ | $5,254 \pm 467$ | $4,923 \pm 298$ | $\mathbf{5,583} \pm 328$ | $5,375 \pm 211$ | $5,429 \pm 125$ |
| Swimmer | $243 \pm 128$ | $348 \pm 2$ | $265 \pm 117$ | $314 \pm 87$ | $349 \pm 24$ | $347 \pm 9$ | $\mathbf{351} \pm 5$ |
| *Fetch (v4)* | | | | | | | |
| Push | $0.9 \pm 0.35$ | $\mathbf{0.99} \pm 0.1$ | $0.8 \pm 0.4$ | - | $\mathbf{0.99} \pm 0.1$ | $0.99 \pm 0.11$ | $\mathbf{0.99} \pm 0.1$ |
| Slide | $0.62 \pm 0.15$ | $0.78 \pm 0.34$ | $0.66 \pm 0.19$ | - | $0.91 \pm 0.24$ | $0.74 \pm 0.26$ | $\mathbf{0.85} \pm 0.25$ |
| PickAndPlace | $0.65 \pm 0.18$ | $\mathbf{0.99} \pm 0.1$ | $0.81 \pm 0.27$ | - | $\mathbf{0.99} \pm 0.05$ | $0.77 \pm 0.13$ | $0.78 \pm 0.21$ |

**Robustness.** We conduct additional experiments on Gym-v4 (Appendix B.7) to further validate the robustness of our default hyperparameter choices. These additional experiments confirm the reliability and effectiveness of our proposed configurations for AEAP under various experimental conditions.

## 5.2 Ablation Studies on Main Hyperparameters

Figure 8 presents ablation results on the sensitivity of initial actor count $N \in \{3, 5, 7\}$ and the distance pruning threshold coefficient $c \in \{0.5, 1.0, 1.5\}$ using the Walker2d and Humanoid as representative intermediate and high dimensional environments. The first row shows training curves and the second row tracks how the number of actors evolves throughout training. For direct comparison with single-actor methods, we include a black line indicating the mean performance of TD3. Several key observations are revealed:

- **Small** $c = 0.5$**:** Minimal pruning occurs (first column in 8a and 8b), with actor counts remaining near initial values throughout training and training curves intertwine without clear differentiation. This matches our expectations: without explicit regularization, randomly selecting among multiple actors resembles running multiple independent TD3 instances. While actors show no performance stratification, initial learning accelerates compared to standard TD3 before eventually converging to baseline performance, consistent with our exploration analysis in Section 4.2.

- **Moderate** $c = 1.0$**:** Actor counts reduce to approximately half the initial ensemble size (second column in 8a and 8b), with clear performance stratification emerging. Smaller ensembles ($N = 3$ or $N = 5$) tend to yield superior performance than the larger one ($N = 7$) because the dual-randomized update scheme gives each actor fewer gradient steps in larger ensembles, slightly hampering individual learning. Nevertheless, even with reduced update frequency, pruned ensembles outperform both unpruned ensembles and single-actor baselines, demonstrating the value of selective retention over fixed-size approaches.

- **Large** $c = 1.5$**:** Aggressive pruning rapidly converges to $1 - 2$ actors after initial exploration (third column in 8a and 8b). Larger ensembles (e.g., $N = 5$ or $N = 7$) achieve superior final performance by fully exploiting multi-actor exploration before consolidation, while smaller ensembles $N = 3$ suffer from insufficient early diversity due to the premature aggressive pruning.

- **Pruning dynamics:** Larger ensembles consistently prune faster across all $\zeta$ values. Despite orthogonal initialization, more actors naturally lead to greater action-space overlap, triggering distance-based pruning more frequently.

It is important to note that the heightened sensitivity observed in Figure 8 results from deliberately testing boundary conditions with exaggerated pruning thresholds rather than reflecting realistic deployment scenarios.

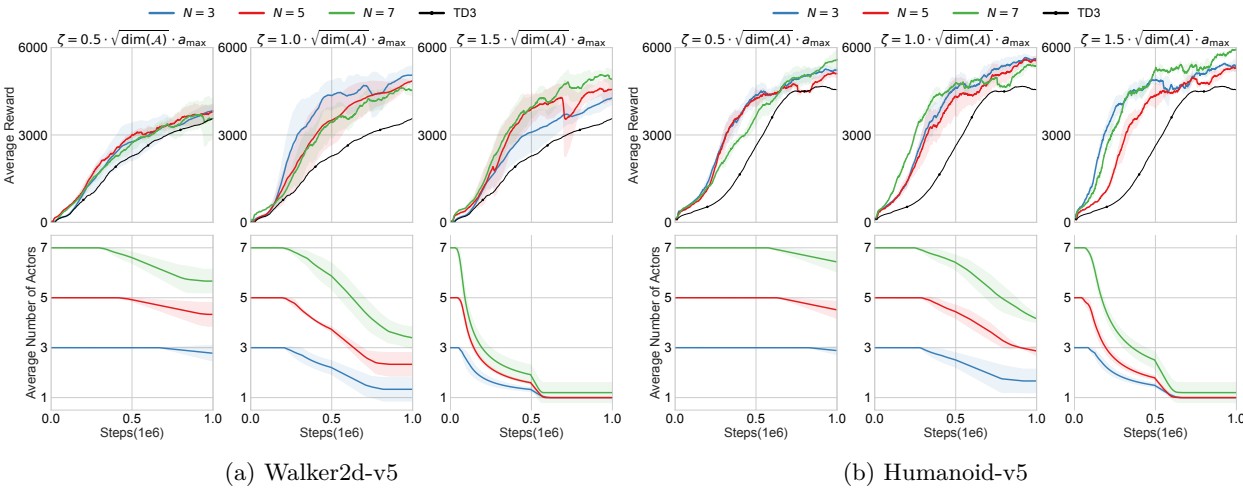

Figure 8: Ablation studies on main hyperparameters. Top rows show training performance (average reward) and bottom rows track ensemble size evolution (average number of actors) across different initial actor counts $N$ and distance thresholds $\zeta$. With small $\zeta$, minimal pruning occurs and performance remains close to TD3 baseline. Moderate $\zeta$ enables effective pruning with clear performance improvements, while aggressive $\zeta$ rapidly reduces ensemble size, with larger initial ensembles achieving superior final performance.

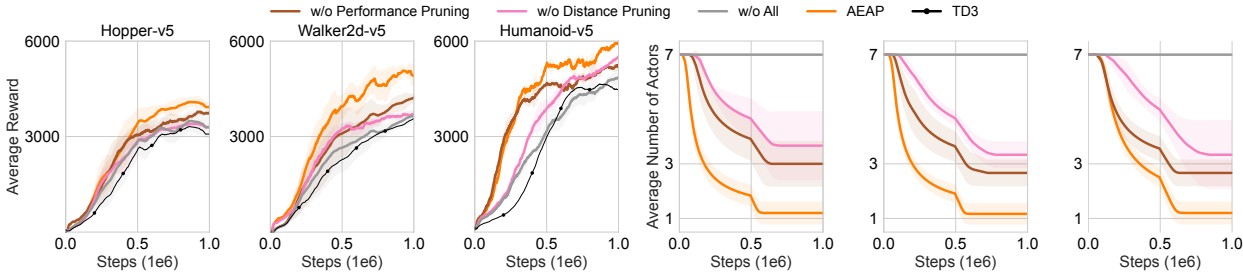

Figure 9: Ablation studies on pruning mechanisms, showing that combined pruning achieves the highest performance.

We provide hyperparameter selection guidelines below that are designed to be domain-agnostic. Similar trends are observed across additional domains, as detailed in Appendix B.8.

**Hyperparameter selection guidelines.** Hyperparameter selection can be distilled into two scenarios: (1) adopt the conservative setting when sustained behavioural diversity is valuable, as in non-stationary, multi-objective, or risk-sensitive domains where multiple distinct policies hedge against regime shifts; and (2) choose the aggressive setting when the aim is to produce one high-performing policy quickly, as in benchmark control suites or fixed industrial tasks, leveraging broad early exploration followed by rapid, compute-efficient convergence.

## 5.3   Ablation Studies on Pruning Mechanisms

We ablate the two pruning mechanisms independently using the aggressive configuration ($N = 7$, $\zeta = 1.5 \cdot \sqrt{\dim(\mathcal{A})} \cdot a_{\max}$) across three environments of increasing action dimensionality: Hopper (3-D), Walker2d (6-D), Humanoid (17-D). Figure 9 reveals that combined pruning achieves the highest performance and most efficient actor reduction, validating that both mechanisms are essential and complementary. Distance-based pruning is triggered more frequently than performance-based pruning across all environments, indicating that action-space similarity is more common than performance differences during training.

## 6 Discussion and Limitations

This paper presents Actor Ensemble with Adaptive Pruning (AEAP), a novel approach that addresses the fundamental tension between exploration diversity and computational efficiency in ensemble-based reinforcement learning. Our key contributions include: (1) demonstrating that dual-randomized actor selection naturally maintains behavioral diversity without explicit regularization terms, (2) developing an adaptive dual-criterion pruning mechanism that evaluates actors based on both critic-estimated performance and action-space similarity, enabling the ensemble to harness early diversity for exploration while progressively eliminating redundant or underperforming actors for computational efficiency, and (3) showing that AEAP achieves lower computational overhead compared to fixed-size ensembles like Gamid (Dey & Sharon, 2024) and PMOE (Ren et al., 2021) while maintaining superior performance, particularly in sparse-reward environments where exploration is crucial. By automatically adjusting ensemble size based on the learning phase, maintaining diversity during exploration and consolidating during exploitation, AEAP provides a practical framework for deploying ensemble methods without the computational burden of maintaining redundant actors throughout training.

Although our experiments illustrate the flexibility of this approach, several limitations warrant further investigation. First, AEAP introduces four additional hyperparameters compared to TD3. Two of these, performance threshold $\xi = 0.85$ and pruning frequency $\kappa = 10,000$, perform robustly across all tested domains without requiring per-task tuning. However, Section 5.2 reveals that the scaling factor $c$ and initial ensemble size $N$ exhibit sensitivity under extreme settings. Though our ablation studies identify two robust configurations, conservative pruning ($N{=}3, c{=}1.0$) and aggressive pruning ($N{=}7, c{=}1.5$), that transfer reliably across all nine domains, we acknowledge that optimal hyperparameter selection may benefit from domain-specific adaptation in certain scenarios.

Second, formal convergence guarantees on how pruning affects convergence rates and sample complexity remain an open problem. Gal & Ghahramani (2016) prove dropout approximates variational inference, and Osband et al. (2016) demonstrate bootstrapped ensembles implement Thompson sampling by maintaining diverse posterior samples over value functions. Our random actor selection similarly prevents deterministic co-adaptation, preserving epistemic uncertainty throughout training. Additionally, delayed updates from randomly selecting actors for backpropagation align with asynchronous optimization theory, where Zinkevich et al. (2010) show parallel actors with gradient delays achieve linear speedup, and Osband et al. (2018) prove asynchronous SGD retains convergence when delays remain bounded. Each actor in AEAP creates inherent delays that facilitate escape from sharp local minima, as observed in our bandit experiments (Figure 4). However, existing analyses assume tabular settings or convex objectives, whereas AEAP operates in high-dimensional continuous control with non-convex policy networks and adaptive ensemble sizes. Extending formal guarantees to this setting remains a challenging problem.

We believe this work nevertheless opens many exciting directions for future research. First, theoretical analysis of convergence guarantees under adaptive ensemble sizes would strengthen the foundation of this approach and guide principled hyperparameter selection. Second, investigating adaptive pruning criteria that incorporate uncertainty estimates or model-based predictions could further improve efficiency. We hope this work inspires further exploration of adaptive ensemble methods that balance exploration, exploitation, and computational efficiency in deep reinforcement learning.

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

# A Additional Related Work

**Exploration in Single-Agent RL.** Exploration allows agents to visit under-explored regions of the environment by deviating from optimal policies. Many approaches induce exploration by introducing stochasticity, either by adding noise directly to the output actions (Fujimoto et al., 2018; Mnih et al., 2015; Lillicrap et al., 2019) or to the policy parameters (Plappert et al., 2018b; Fortunato et al., 2019). Other strategies explicitly optimize for exploration, such as maximizing action entropy (Haarnoja et al., 2017; 2018) or maintaining visitation maps that motivate visits to infrequently encountered states (Tang et al., 2017; Ostrovski et al., 2017). Intrinsic rewards (Burda et al., 2018b; Stadie et al., 2015; Dey et al., 2024) and curiosity-driven objectives (Burda et al., 2018a; Pathak et al., 2017) have also been utilized. However, these methods fundamentally rely on modifying or augmenting the agent's current policy, thus inherently limiting exploration scope based on previously learned behaviors.

**Critic Ensembles in RL.** Another prominent line of research maintains multiple critic networks (value functions) to guide exploration more effectively. Osband et al. (2016) trains multiple Q-heads to approximate Thompson sampling and Anschel et al. (2017) reduces variance by averaging historical predictions. Osband et al. (2018) injects fixed bias to preserve long-term uncertainty, Lan et al. (2020) mitigates overestimation via conservative ensemble targets. Lee et al. (2021) adds an uncertainty bonus to the ensemble for improved exploration. Chen et al. (2021) extends these ideas to continuous control by updating a ten-critic ensemble with random subsampling, achieving state-of-the-art sample efficiency.

# B Additional Experimental Results

This section provides comprehensive experimental results to support the findings presented in the main text.

## B.1 Exploration Effectiveness in Low-Dimensional Environments for TD3

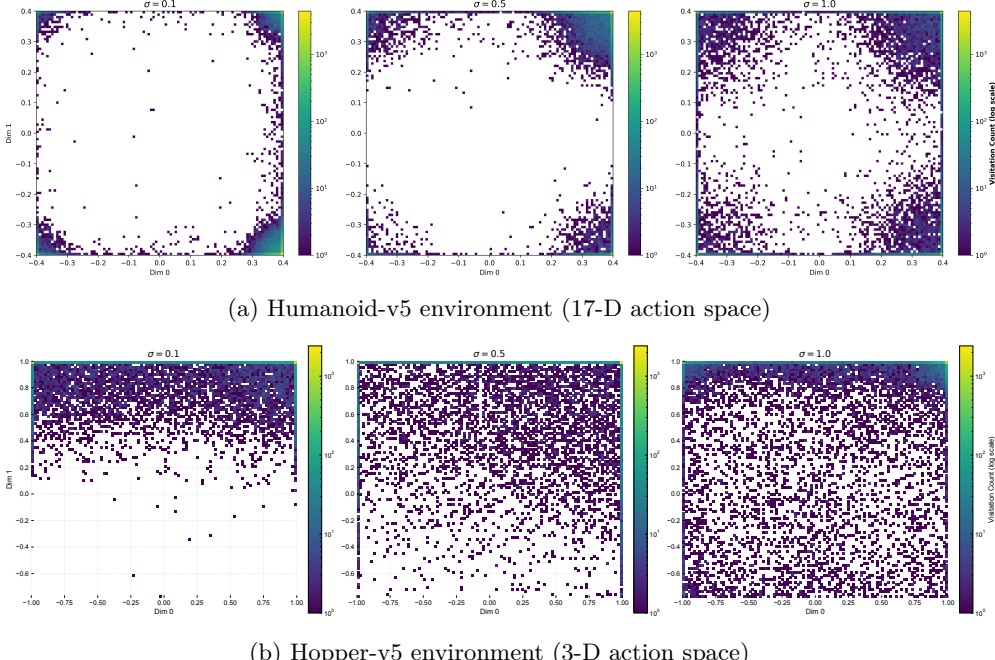

(a) Humanoid-v5 environment (17-D action space)

(b) Hopper-v5 environment (3-D action space)

Figure 10: Action distribution heatmaps showing coverage across the first two action dimensions for TD3 with varying exploration noise levels.

Figure 10.(b) presents action distribution heatmaps for TD3 in the Hopper environment. Unlike the high-dimensional Humanoid environment, increasing exploration noise in Hopper demonstrably improves action

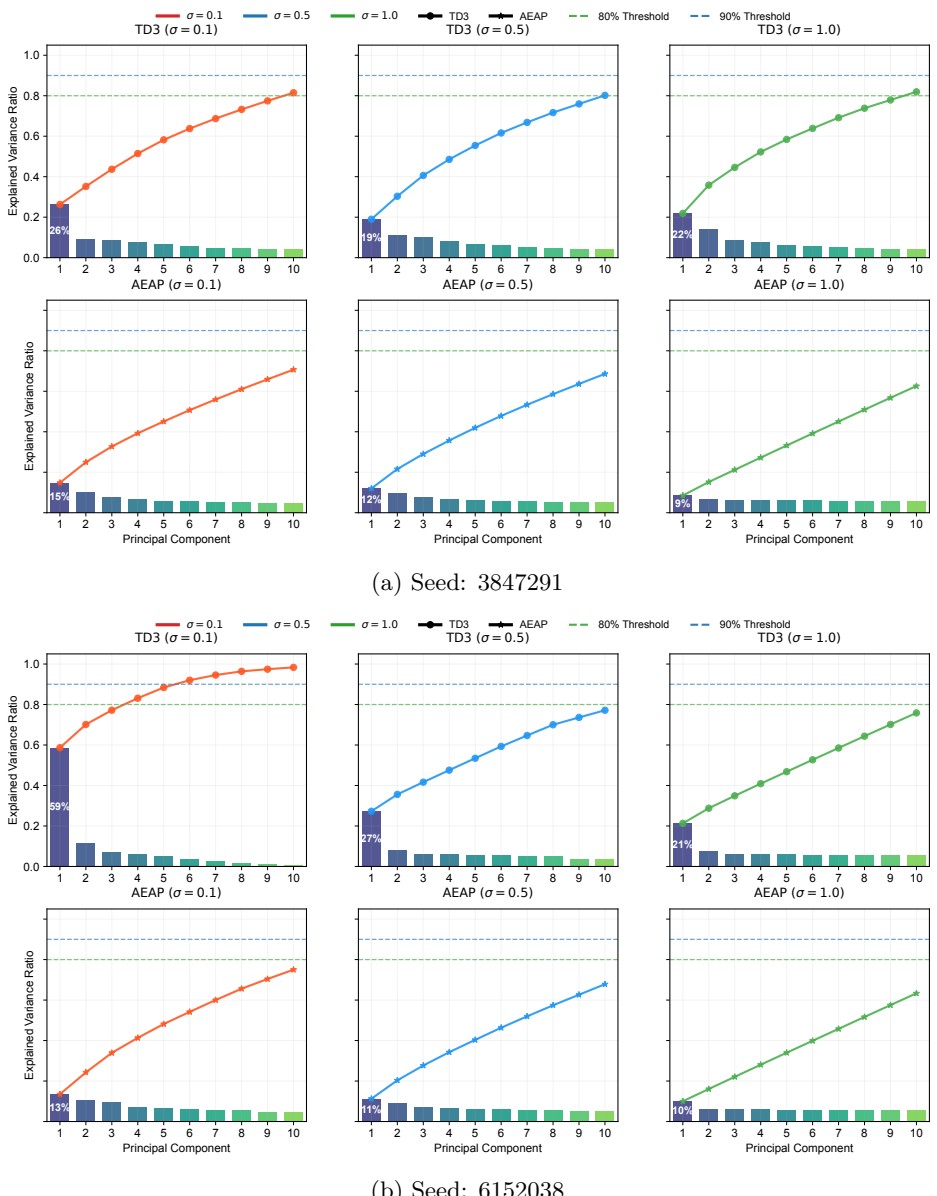

Figure 11: PCA analysis in Humanoid-v5 environment across different random seeds.

coverage across the visualized dimensions. This observation corroborates the dimensionality-dependent nature of exploration effectiveness in single-actor methods.

## B.2 Exploration Effectiveness in High-Dimensional Environments for TD3

We provide additional empirical evidence supporting our claim from Section 4.2 that increasing exploration variance in single-actor methods is insufficient for robust exploration in high-dimensional action spaces.

Figures 11 presents PCA projections of TD3's 17-dimensional action outputs in the Humanoid environment across another two random seeds, with exploration noise $\sigma \in \{0.1, 0.5, 1.0\}$. These results demonstrate the consistency of TD3's limited exploration patterns across different initializations, complementing the analysis presented in Figure 3 of the main text.

### B.3 1-D Continuous Bandit Experimental Setup

The environment features a 1-dimensional action space $a \in [-1, 1]$ with a deterministic bimodal reward function. The reward landscape contains a narrow global optimum at $a = -0.7$ (reward $= 1.0$) and a wide local optimum at $a = 0.3$ (reward $= 0.5$), as illustrated in Figure 12. This design creates an exploration-exploitation trade-off where the global peak is challenging to discover due to its narrow width, while the local peak is easily found but yields suboptimal rewards.

In the all-actor update configuration, all actors in the ensemble receive gradient updates simultaneously using the same batch of experiences, contrasting with the single-actor update where only one randomly chosen actor is updated per step. Updates occur at every timestep with a batch size of 32.

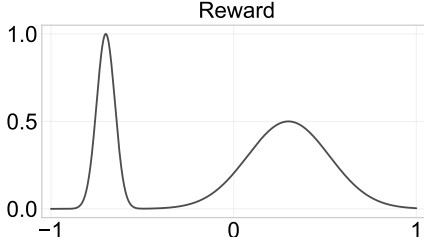

Figure 12: 1-D Continuous Bandit environment

Figure 13 tracks cumulative action selection probabilities over time for three actors, revealing distinct convergence patterns.

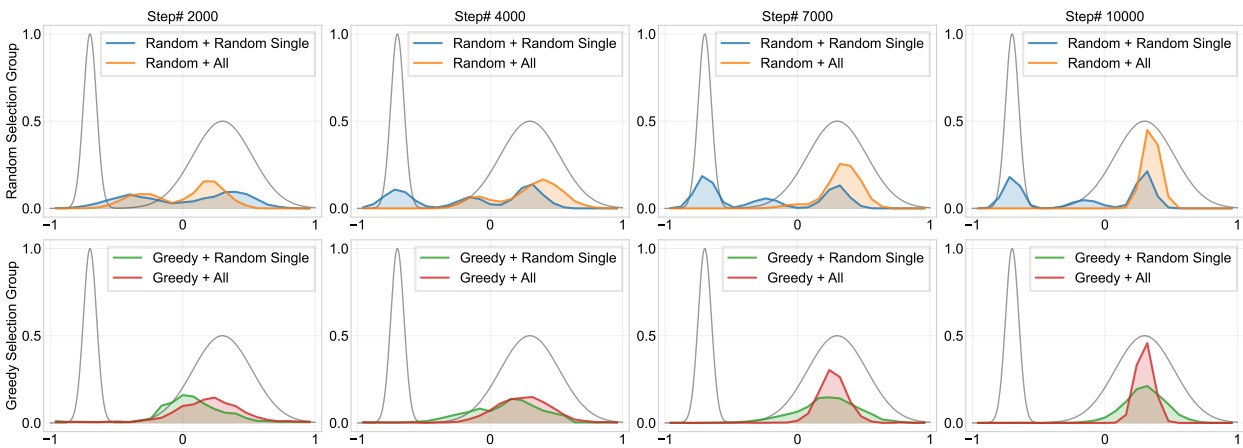

Figure 13: Cumulative action selection probabilities under four actor-management schemes with three actors. Random-Single maintains diversity but also trapped between optimal peaks.

### B.4 Running Statistics Normalization (RSNorm)

We adopt this normalization technique from (Lee et al., 2025), which demonstrates its superior performance compared to alternative normalization methods. RSNorm standardizes input observations by tracking the running mean and variance of each input dimension during training, preventing features with disproportionately large values from dominating the learning process.

Given an input observation $\mathbf{o}_t \in \mathbb{R}^{d_o}$ at timestep $t$, we update the running observation mean $\mu_t \in \mathbb{R}^{d_o}$ and variance $\sigma_t^2 \in \mathbb{R}^{d_o}$ as follows:

$$\mu_t = \mu_{t-1} + \frac{1}{t}\delta_t, \quad \sigma_t^2 = \frac{t-1}{t}\left(\sigma_{t-1}^2 + \frac{1}{t}\delta_t^2\right) \tag{1}$$

where $\delta_t = \mathbf{o}_t - \mu_{t-1}$ and $d_o$ denotes the dimension of the observation.

Once $\mu_t$ and $\sigma_t^2$ are computed, each input observation $\mathbf{o}_t$ is normalized as:

$$\tilde{\mathbf{o}}_t = \text{RSNorm}(\mathbf{o}_t) = \frac{\mathbf{o}_t - \mu_t}{\sqrt{\sigma_t^2 + \varepsilon}} \tag{2}$$

where $\tilde{\mathbf{o}}_t \in \mathbb{R}^{d_o}$ is the normalized output, and $\varepsilon$ is a small constant for numerical stability.

In our case, after applying RSNorm, we use `np.digitize` to discretize the state space. For Walker2d, we discretize the 17-dimensional observation space into 10 bins, and for Humanoid, we discretize the 347-dimensional observation space into 50 bins.

### B.5 Overestimation Bias in Pruning Decisions

A potential concern with performance-based pruning is whether overestimation bias in $Q$-networks could lead to incorrect pruning decisions, where actors with better value estimates are mistakenly removed due to critic inaccuracies.

We do not observe this situation in experiments, probably due to several mitigating factors. First, we inherit double $Q$-learning mechanism, which addresses overestimation bias by taking the minimum of two critic estimates. Second, we use average $Q$-values over entire batches, which naturally smooths out extreme outliers — only actors that consistently underperform get identified for pruning. Third, during the initial freeze period where both actors and critics have high variance, no pruning occurs. After this period, our pruning frequency is every 10,000 steps, which is conservative and allows critics to mature.

A particularly influential parameter may be batch size, as smaller batches introduce higher variance in $Q$-value estimation, potentially causing the pruning issues mentioned above. The following table shows experiments across different batch sizes yield similar performance across several domains (5 runs):

| Environment | Batch Size | | |
| --- | --- | --- | --- |
| | 256 | 512 | 1024 |
| Walker2d-v5 | 5,209±467 | 5,181±287 | 5,173±94 |
| Humanoid-v5 | 5,429±125 | 5,552±198 | 5,341±186 |
| Ant-v5 | 5,632±199 | 5,462±191 | 5,648±103 |

Table 3: Performance consistency across batch sizes (5 runs)

### B.6 Integration of Pruning Strategies with GAMID and PMOE

Figure 14 shows that integrating pruning strategies improves performance for both GAMID and PMOE in HalfCheetah and Humanoid. The largest gains occur when both redundancy-based and performance-based criteria are combined.

### B.7 Learning Curves of AEAP on Gym-v4 and Gym-v5

Figure 16 demonstrates the performance of AEAP on MuJoCo-v5 environments under the aggresive parameter configuration: $N = 7$ actors with distance threshold $\zeta = 1.5 \cdot \sqrt{\dim(\mathcal{A})} \cdot a_{\max}$.

Figure 16 demonstrates the performance of AEAP on MuJoCo-v4 environments under the conservative parameter configuration: $N = 3$ actors with distance threshold $\zeta = 1.0 \cdot \sqrt{\dim(\mathcal{A})} \cdot a_{\max}$. The results confirm

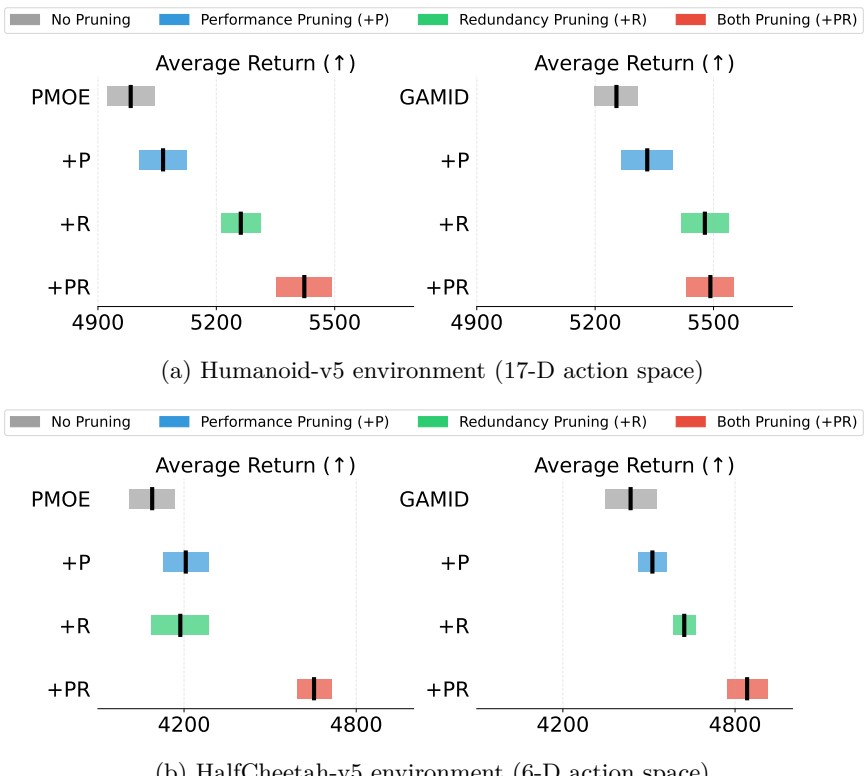

(a) Humanoid-v5 environment (17-D action space)

(b) HalfCheetah-v5 environment (6-D action space)

Figure 14: Integration of pruning strategies with GAMID and PMOE on different Mujoco tasks.

that AEAP consistently outperforms or matches baseline methods across different Gym environment versions, establishing the robustness of our approach to environmental variations.

### B.8    Additional Ablation Studies on Main Hyperparameters

Following standard practice in (Haarnoja et al., 2018), we provide comprehensive sensitivity analyses across all MuJoCo environments. Figures 17 through 19 extend the hyperparameter analysis from Section 5.2, presenting detailed ablation studies on initial actor count $N$ and distance pruning threshold $c$ for each environment. These results validate the robustness of our proposed configurations and confirm the parameter selection guidelines across diverse continuous control tasks.

## C    Hyperparameters And Compute

We used the tuned hyperparameter values for SAC and TD3, that have been provided in RL Baselines3 Zoo which is built on top of Raffin et al. (2021). For Gamid-PG, we used hyperparameters recommended in Dey & Sharon (2024). For SimBa, we used recommended hyperparameters from Lee et al. (2025) with minor adjustments.

All reported experiments were distributed between 3 machines; (1) a machine with 64 32-core AMD Ryzen Threadripper PRO 5975WX CPUs, each clocked at 4.3 GHz with 250 GB RAM with 2 NVIDIA GeForce RTX 3090 24 GB GPUs (2) a machine with an Intel(R) Core(TM) i9-13900K CPU with 24 cores (8 performance + 16 efficiency cores), base clock at 3.0 GHz (boost up to 5.8 GHz) with 32 GB RAM and an NVIDIA GeForce RTX 4090 24 GB GPU.

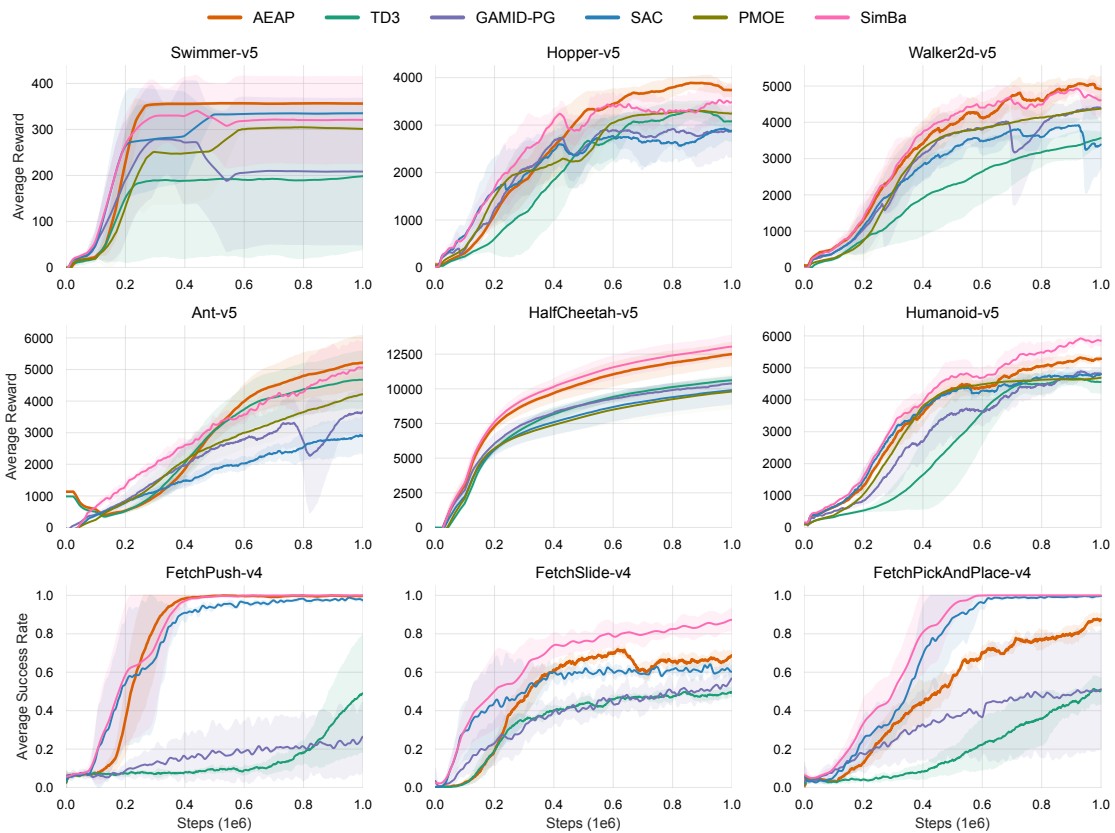

Figure 15: Training performance on MuJoCo-v5 environments. Shaded regions represent standard deviation across 7 independent trials. Curves are smoothed using a 100-step moving window for clarity.

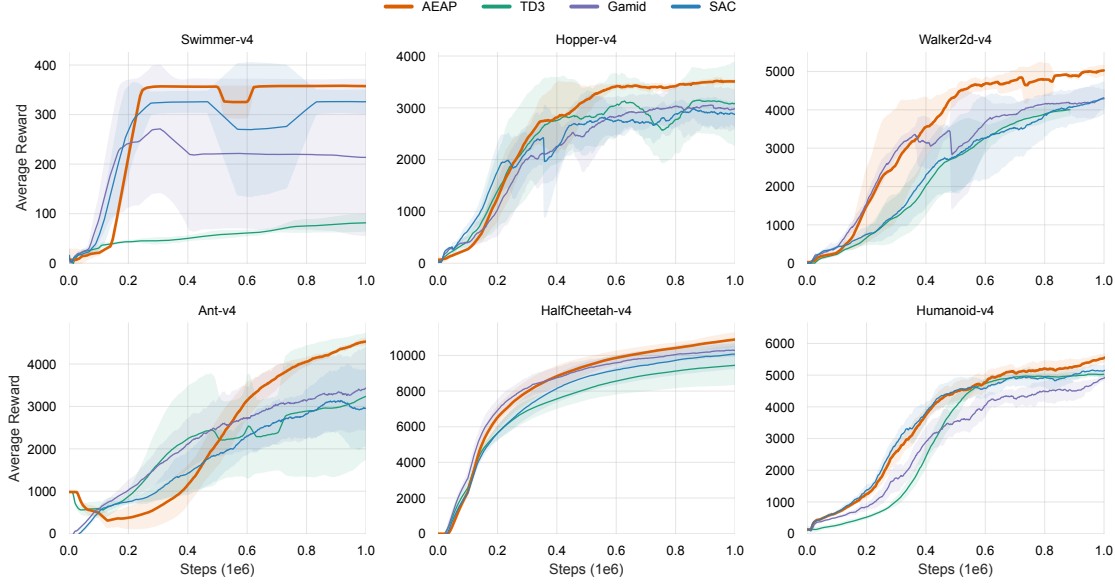

Figure 16: Training performance on MuJoCo-v4 environments. Shaded regions represent standard deviation across 7 independent trials. Curves are smoothed using a 100-step moving window for clarity.

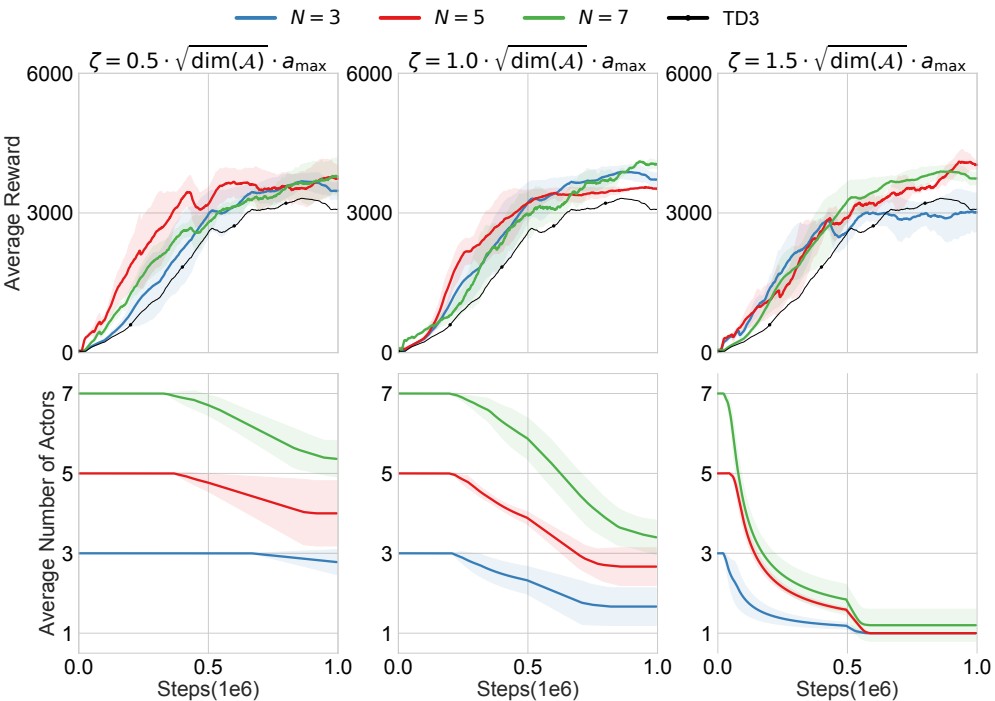

Figure 17: Hyperparameter ablation results for Hopper-v5 environment.

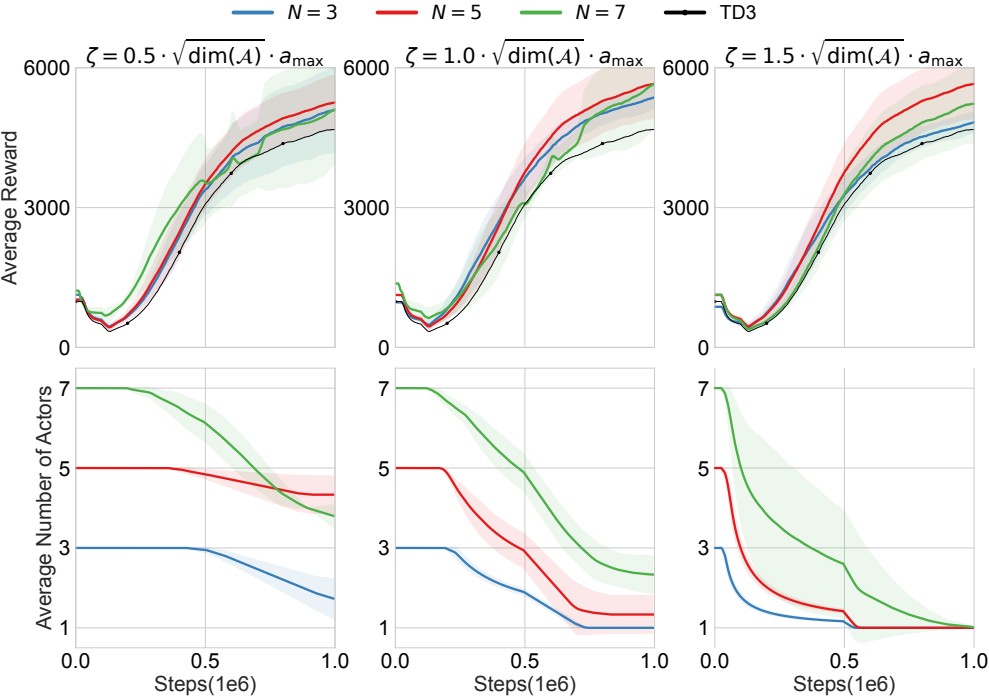

Figure 18: Hyperparameter ablation results for Ant-v5 environment.

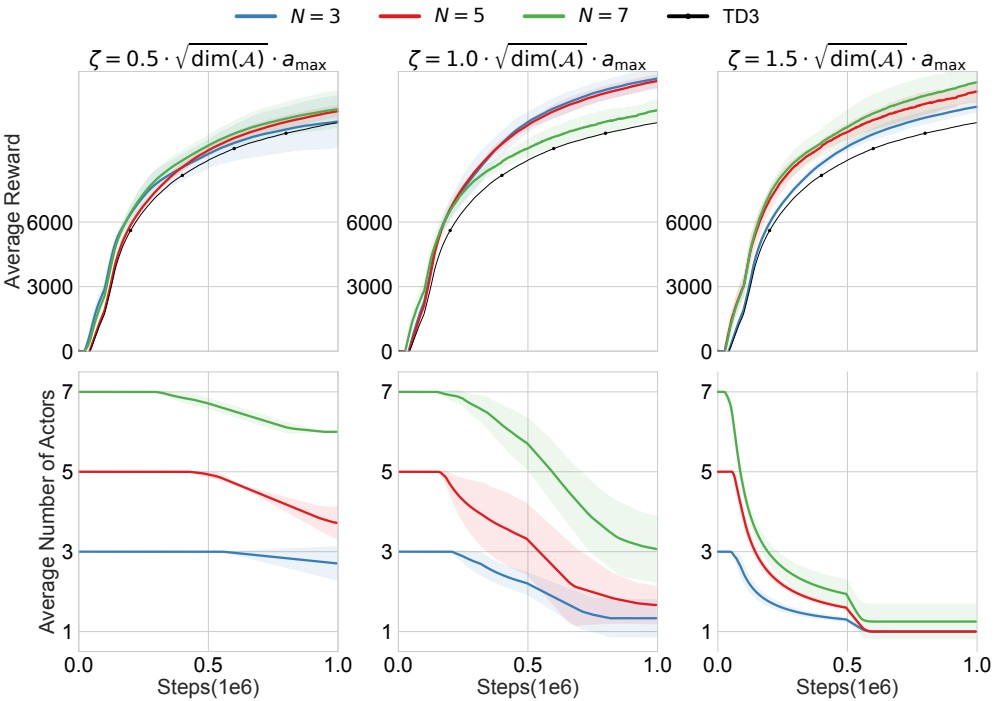

Figure 19: Hyperparameter ablation results for HalfCheetah-v5 environment.

Table 4: SimBa hyperparameters.

| Hyperparameter | Value |
|---|---|
| Critic block type | SimBa Residual |
| Critic num blocks | 2 |
| Critic hidden dim | 512 |
| Critic learning rate | 1e-4 |
| Target critic momentum ($\tau$) | 5e-3 $\rightarrow$ 1e-3 |
| Actor block type | SimBa Residual |
| Actor num blocks | 1 |
| Actor hidden dim | 128 |
| Actor learning rate | 1e-4 $\rightarrow$ 3e-4 |
| Initial temperature ($\alpha_0$) | 1e-2 |
| Temperature learning rate | 1e-4 |
| Target entropy ($\mathcal{H}^*$) | $|\mathcal{A}|/2$ |
| Batch size | 256 |
| Optimizer | AdamW |
| Optimizer momentum ($\beta_1, \beta_2$) | (0.9, 0.999) |
| Weight decay ($\lambda$) | 1e-2 |
| Discount ($\gamma$) | Heuristic |
| Replay ratio | 2 $\rightarrow$ 1 |
| Clipped Double Q | HumanoidBench: True
Other Envs: False |

