# OpenReview forum: "AEAP: A Reinforcement Learning Actor Ensemble Algorithm with Adaptive Pruning"
_TMLR — Accepted by TMLR_

### Review · Reviewer_vsrG · 2025-08-04

**Summary Of Contributions:**

The paper proposes a novel algorithm for actor ensembles in continuous control reinforcement learning. The core contributions are the introduction of a "dual-randomized" strategy, in which a random actor is chosen to execute actions in the environment, and independently an actor is chose which receives a policy update. In addition, the paper proposes actor pruning as actors either collapse to the same policy over time, or lag behind in performance.

The paper presents a mostly empirical, and thorough, evaluations of the proposed method without theoretical investigation.

**Additional Comments:**

n/a

**Audience:**

Yes

**Audience Explanation:**

The paper addresses an important question, effectively training ensembles in off-policy actor critic settings.

**Claims And Evidence:**

Yes

**Claims Explanation:**

The claimed contributions seem mostly supported by the empirical evidence. Overall, the ideas are sound, however, their impact on performance as well as supporting evidence for the claims is somewhat thin.

This is mostly due to three issues:
* The paper proposes two orthogonal improvements to actor-ensemble training, which do not seem to be related in an intuitive manner.
* The proposed method for ensuring diversity is heuristic in nature.
* The paper evaluates baselines and other methods with strongly outdated architectures.

The first issue is hopefully somewhat self explanatory. I think that it would be useful to discuss pruning when combined with other strategies for obtaining diverse expert actors, such as those presented by the baselines. The idea seems to be readily combineable with other.

The second issue arises because the mechanisms which underlie the dual-actor selection method are not discussed or evaluated in detail. While a theoretical investigation would be great, but is potentially out-of-scope for the paper, I think it would greatly benefit the community if the work sought to clarify why the proposed strategy leads to multimodal policies. In the context of replay-buffer based off-policy learning, all actors will randomly see all data at some point, and so it is not quite clear whether the proposed strategy does indeed lead to stable diversity, or simply delays the updates of all actors.

The third issue arises as the authors seem to only test their method with outdated architectures for TD3 and SAC. A lot of recent work [1,2,3,4] has focused on producing more reliable learning in actor-critic methods, and it should be evaluated if the strategies proposed here robustly generalize to state-of-the-art approaches. This is highly relevant as the much more stable value learning of these approaches might necessitate explicit regularization to ensure actor diversity.

While it has sadly been a standard in RL to base advancements on highly outdated algorithms and architectures such as SAC and TD3 in their original formulation, I have decided to become more strict in admonishing this common practice in my reviews. I ask the AC to intervene in case they believe this is take into consideration that this is a harsher standard than other reviewers might apply.

[1] Bigger, Regularized, Optimistic: scaling for compute and sample-efficient continuous control, Nauman et al., Neurips 2024, https://sites.google.com/view/bro-agent/
[2] SimBa: Simplicity Bias for Scaling Up Parameters in Deep Reinforcement Learning, Lee et al., https://openreview.net/forum?id=jXLiDKsuDo
[3] Dissecting Deep RL with High Update Ratios: Combatting Value Divergence, Hussing et al., https://openreview.net/forum?id=ofwv9VYp3h
[4] CrossQ: Batch Normalization in Deep Reinforcement Learning for Greater Sample Efficiency and Simplicity, Bhatt et al., https://openreview.net/forum?id=PczQtTsTIX

**Requested Changes:**

The ordering of the sections in the paper is slightly suboptimal. It would be preferable to have a fully specified algorithm (Section 4.3) before showing experiments (Section 4.1). Otherwise the reader is left wondering what exactly is being evaluated.

Figure 5 is deeply confusing, I think it would be much clearer to show performance over time, or time to reach some set reward in a standard bar plot, than merely describing the time to finish the experiments. In addition, the complete lack of scale on the graph makes it seem arbitrary and like a poor choice of visualization.

Minor typos:

4.3 Actor Ensmble with Adaptive Pruning > Actor Ensemble

---

### Review · Reviewer_nf1p · 2025-08-18

**Summary Of Contributions:**

This paper introduces a new reinforcement learning algorithm, AEAP (Actor  Ensemble with Adaptive Pruning), which addresses the challenge of intractable policy  gradients often encountered in ensemble methods using stochastic policies. By  adopting deterministic policies, the proposed method improves gradient tractability  and introduces two key innovations: (1) Dual-Randomized Actor Selection: This mechanism decouples exploration and  learning by employing different random actor selections for environment interaction  and policy updates. It effectively maintains policy diversity without requiring explicit  regularization terms. (2) Actor Pruning: Actors are progressively pruned based on two complementary criteria, namely estimated value from the critic and action-space similarity, allowing the framework to eliminate underperforming or redundant actors dynamically. Extensive experiments on both MuJoCo and Fetch benchmark tasks demonstrate AEAP's competitive performance and computational efficiency compared to the existing baselines.

**Strengths**

1. The proposed algorithm is conceptually simple and easy to follow. By presenting  experimental analysis to motivate the design choices, the paper illustrates how each modification contributes to the performance improvement.

2. The experiments are quite thorough, demonstrating that AEAP consistently  outperforms other deterministic methods (TD3, Gamid-PG) across most environments. Regarding training efficiency, AEAP also surpasses other multi-actor ensemble approaches. Additionally,  the paper includes ablation studies on the number of actors and the pruning threshold, further validating the effectiveness of the proposed design.

3. As an extension of Gamid, AEAP addresses the limitations of actor ensembles in  sparse reward tasks, where performance often degrades. Moreover, it achieves this  with lower computational overhead.

**Weaknesses**

1. Although the paper states that the motivation is to address the difficulty of tuning
regularization parameters, the proposed method introduces several new pruning related hyperparameters. While two general configurations are provided, the  results still show sensitivity to different tasks and environments. Therefore, the  approach appears less attractive as a solution for improving algorithmic stability.

2. The paper focuses primarily on practical implementation improvements, but lacks  theoretical grounding to support the proposed method. Specifically, it remains somewhat unclear why dual-randomized actor selection can in general encourage exploration. I can see that the proposed approach appears to work well in the 1D toy example (cf. Figure 2), but this result does not imply that this holds for RL problems in general. More justifications are needed to support that this design is indeed principled.

3. Based on Table 2, it seems that the performance of AEAP is statistically about the same as the vanilla SAC in most of the tasks (e.g., Hopper, Humanoid, Halfcheetah, Swimmer, and the three Fetch tasks). Therefore, this makes AEAP not that appealing as simple SAC can already achieve similar total returns.

4. Some of the discussions on actor ensemble need further explanations:
- Page 2, first paragraph: It is stated that “when treating stochastic ensemble policies as components of a probabilistic mixture, computing exact policy gradients becomes analytically intractable, requiring surrogate approximations that can compromise performance.” However, in typical RL, one does not need to compute the policy gradient exactly through an analytical form. Instead, a stochastic estimate of the gradient is sufficient. For example, even if the action distribution follows a mixture Gaussian, an estimate of the policy gradient can still be calculated as long as the score function $\nabla \log\pi(a\rvert s)$ can be calculated numerically. Therefore, I do not see why this is actually a challenge.
- Page 4, third paragraph of Section 4.2: It is stated that “A plausible explanation is that, maintaining several actors benefits exploration in the early phase when state–action visitation is sparse. However, as exploration becomes less valuable relative to exploitation, higher $\epsilon$ enforces more random selections when the algorithm should exploit the superior actor, and thus leads to worse performance.” As Table 1 only shows the final action selection ratios and final mean performance under different $\epsilon$, the result does not provide much information about the training dynamics. Hence, it is unclear why Table 1 suggests that pruning is helpful at a later training stage.

**Audience:**

Yes

**Audience Explanation:**

The ideas of dual-randomized actor selection and pruning mechanisms are likely to be of interest to TMLR’s audience, particularly those researching ensemble-based RL or focused on improving exploration, despite that AEAP does not always achieve the best performance across all environments.

**Claims And Evidence:**

Yes

**Claims Explanation:**

For both proposed enhancements, the authors provide empirical evidence  demonstrating that the method effectively enhances exploration without requiring  additional regularization terms, while also reducing computational cost. AEAP also  clearly outperforms other ensemble RL methods in sparse reward settings. However, as noted in the weaknesses, it remains unconvincing that replacing regularization with pruning-based hyperparameters leads to a more stable or generally robust solution.

**Requested Changes:**

1. Including theoretically-grounded explanation or analysis and additional empirical evidence demonstrating that the proposed pruning strategy is more robust than regularization-based approaches.

2. Currently, the paper primarily highlights how AEAP improves the exploration  issue in TD3. It would be beneficial to provide further explanation on why AEAP outperforms other actor ensemble methods specifically in sparse reward environments.

3. To make sure that the improvement of AEAP over the baselines is indeed statistically significant across tasks, some reliable metrics (e.g., interquartile mean) can also be provided to strengthen the empirical results.

---

### Review · Reviewer_ogz1 · 2025-09-04

**Summary Of Contributions:**

The authors propose a new RL algorithm Actor Ensemble with Adaptive Pruning (AEAP), an ensemble-based method that shows strong performance while also being computationally efficient. AEAP combines randomized actor selection with adaptive pruning to maintain diversity and improve efficiency. Extensive experiments and analysis are provided.

**Additional Comments:**

Overall a well-written paper, I appreciate the extensive analysis and experiments.

Downsides: the additional complexity is not ideal, the performance improvement alone is also not that strong compared to the baselines. Though a combination of performance and efficiency are good.

Minor issue: can always benefit from even more experiments on harder benchmarks. Though I acknowledge the current experiments are quite extensive.

Question: if we do not consider computation efficiency and focus on sample efficiency, can the performance of the proposed method be further improved in terms of sample efficiency?

Question: how well fine-tuned are the baseline methods on some of the fetch environments given they are not typically benchmarked on these? (I believe)

Question: "Actors are progressively removed when their average Q-value falls below 85% of the best performer" interesting way to prune them. Now, what happens if say the Q networks are getting overestimation problems? Could there be cases when the lower Q value ones actually have better estimate of their values and pruning them might be a mistake?

Question: When all training is finished, do you keep a single actor?

**Audience:**

Yes

**Audience Explanation:**

Ensemble-based RL algorithms have been important in improving the performance of RL algorithms, though they can bring a big computation overhead. Studying how to reduce such overhead while maintaining or further improving performance is a fairly important topic. This paper can be especially interesting to people who work in deep reinforcement learning.

**Claims And Evidence:**

Yes

**Claims Explanation:**

The paper provide a quite extensive list of analysis and ablations to help support the claims.

- Figure 4 is interesting and shows Dual-random variant has multimodal distribution even at a later training steps. Figure 3 and 5 also help support the diversity claims.
- Ablations such as in Figure 6 show the effect of each component. Figure 8 shows it is relatively robust to hyperparameters.
- Figure 7 shows the proposed method is quite strong considering both performance and computation efficiency.

Discussions in the paper are quite objective and covered strength of the method but also places where it does not perform the top. Overall I find the claims to be fairly convincing.

**Requested Changes:**

- What are x-axis and y-axis of Figure 4? Please clarify
- It is slightly confusing what the x-axis is in Figure 8 and 9, can you add a bit more caption

---

> ### Author Response · Authors · 2025-09-06
>
> We thank the reviewer for the thoughtful and detailed feedback, and we are pleased that you find our work meaningful. Below we provide point-by-point responses to your questions, with figure revisions detailed at the end.
> ### Q1 - Sample Efficiency
>
> We interpret this question in two ways:
> - (1) whether dual-randomized actor selection or pruning individually improve sample efficiency, and
> - (2) whether the complete AEAP algorithm improves sample efficiency when computational constraints are removed.
>
> For interpretation #1: Our dual-randomized approach promotes exploration diversity, but actors can become trapped in intermediate regions of the action space. As demonstrated in our bandit experiments, while actors discover both optimal and suboptimal regions, they also spend time in less productive intermediate areas. Despite gaining exploration diversity, dual-randomized selection alone doesn't necessarily achieve significant sample efficiency improvements over well-tuned stochastic methods.
>
> Pruning shows different characteristics. *Figure&nbsp;6* demonstrates that integrating pruning with existing ensemble methods increases performance. *Figure&nbsp;9* in our ablation study (Section 5.2) shows steeper learning curves when pruning is used, indicating sample efficiency improvements.
>
> For interpretation #2: The complete method combines both. Learning curves in *Appendix&nbsp;B.6* show our approach demonstrates better sample efficiency than other deterministic policy gradient methods, but performs comparably to stochastic baselines like SAC and SimBa.
> ### Q2 - Tuning on Fetch Environments
> We used established hyperparameter configurations for SAC and TD3 from RL-Baselines3-Zoo, and author-recommended settings for GAMID and PMOE. SimBa is a recently introduced algorithm that we consider the current state-of-the-art baseline(though they just released SimBa v2). I tuned SimBa on FetchSlide for the first 200,000 steps by focusing on different learning rates, and FetchSlide is typically the most challenging one to learn. Then we applied these parameters accorss other Fetch tasks. It seems that stochastic policies still maintain advantages in sparse reward environments, particularly in hierarchical tasks like FetchSlide. We add details on the specifically tuned parameters in *Appendix&nbsp;C*.
> ### Q3 - Overestimation Bias Affecting Pruning Decisions
> This is an interesting concern. We don't observe this situation in experiments, probably due to several mitigating factors. First, we inherit double $Q$-learning mechanism, which addresses overestimation bias by taking the minimum of two critic estimates. Second, we use average $Q$-values over entire batches, which naturally smooths out extreme outliers - only actors that consistently underperform get identified for pruning. Third, during the initial freeze period where both actors and critics have high variance, no pruning occurs. After this period, our pruning frequency is every 10,000 steps, which is conservative and allows critics to mature.
>
> A particularly influential parameter may be batch size, as smaller batches introduce higher variance in estimation, potentially causing the above pruning issues. The following table shows similar performance across different batch sizes on Walker2d (5 runs), and suggests this issue is rare in practice:
> |Batch Size|256|512|1024|
> |---|---|---|---|
> |Avg Return|5,209±467|5,181±287|5,173±94|
>
> We didn't include extensive ablation studies on frequency in the paper since the 10,000 step interval showed good performance, and we wanted to avoid making the algorithm appear overly complex. We include a brief section in *Appendix&nbsp;B.5* covering more domains to clarify this question.
> ### Q4 - Number of Final Actors
> The number of remaining actors depends on the pruning setting used. We can retain more than one actor at the end if the pruning threshold is conservative. *Figure&nbsp;8* tracks the number of actors throughout training and shows the final configuration (we've updated the caption for clarity). During evaluation, we assess all remaining actors and select the best-performing one for deployment.
> ### Changes
> We made the following revisions:
> - We clarified that the background grey curve in *Figure&nbsp;4* represents the bimodal reward landscape in the 1-dimensional action space.
> - We added "Steps (1e6)" as the x-axis label in *Figure&nbsp;8* and provided more detailed descriptions in the figure caption for clarity.

---

> > ### Comment · Reviewer_ogz1 · 2025-09-24
> > **Thank you for the rebuttal**
> >
> > Thank you for the rebuttal, majority of my comments and questions are addressed.

---

### Author Response · Authors · 2025-09-22

Dear AC,

We greatly appreciate the time and effort that you and the reviewers have invested in evaluating our work. We understand that the review process is thorough and takes considerable time, and we are grateful for your patience as our submission moves through the process.
We are happy to provide whatever might be helpful to facilitate the review process, should you need any additional information or clarification.

We look forward to hearing from you when convenient.

---

> ### Comment · Action_Editor_TGFb · 2025-10-21
>
> Dear Authors,
>
> Many thanks for your message! The current status is as follows: we've got all information from the reviewers, and now it is in the decision stage. I expect it would be with you within the next couple of days subject to the approvals.
>
> Best wishes,
> the Action Editor

---

### Decision · Action_Editor_TGFb · 2025-10-20

**Recommendation:** Accept with minor revision

**Additional Comments:**

The authors are expected to outline the limitations  of the analysis in a separate section in a way that addresses the concerns of the reviewers including:
-  a need to tune a number of pruning-related hyper parameters
- limited theoretical justification for the actor selection mechanism.
- clarify upon claims about the policy gradient intractability and pruning benefits, and whether the performance gain upon baselines like SAC are large enough to justify this more complex method.

**Audience:**

Yes

**Audience Explanation:**

The novel ensemble reinforcement-learning methods, proposed in this paper, are of interest to the community, which warrants the positive score on this.

**Claims And Evidence:**

Yes

**Claims Explanation:**

All the reviewers believe that the paper's claims are matched by the evidence, which is a reason for the score.

The reviewers note that the paper introduces two novel techniques:
 (1) Dual-Randomized Actor Selection, which helps maintain policy diversity, and
(2) Actor Pruning to remove underperforming actors.

 The reviewers agree that the strengths of the paper a well-motivated algorithmic design, as well as extensive experiments, interesting empirical results, and clear writing/presentation of the paper.

However, there have been concerns voiced about the paper. This includes a need to tune a number of pruning-related hyper parameters, as well as a limited theoretical justification for the actor selection mechanism, so the results are heavily skewed  upon empirical justification. The reviewers also wanted clarification upon claims about the policy gradient intractability and pruning benefits, and whether the performance gain upon baselines like SAC are large enough to justify this more complex method.

---

> ### Author Response · Authors · 2025-10-30
>
> Dear AC,
>
> Thank you for the constructive feedback and the opportunity to revise our manuscript. We appreciate the reviewers' thorough evaluation. Rather than a separate section, we address all concerns through targeted revisions in the corresponding sections as follows:
>
> * We expand *Section 5.2* to include more detailed discussions on hyperparameter sensitivity and tuning concerns.
> * We strengthen the discussion of our dual-randomized selection mechanism in *Section 4.2*. Specifically, we added more detailed analysis connecting our empirical observations to related theoretical frameworks, while acknowledging the limitations of current theoretical justification.
> * We better enhance *Section 5.1* to articulate AEAP's value proposition. We clarify that while AEAP achieves competitive or superior performance compared to SAC across most benchmarks, its primary advantage lies in reduced wall-clock time, offering practitioners an attractive performance-efficiency trade-off when computational resources or training time are constrained.

---

> > ### Comment · Action_Editor_TGFb · 2025-11-02
> >
> > Dear Authors,
> >
> > Thank you very much for addressing the comments. In my view, such a revision should effectively address the core concerns.
> >
> > However, there is one issue that still needs to be resolved before I can fully assess the updated version and confirm it. It appears that the figures might be missing. For instance, I couldn't locate Figure 1 in the PDF I downloaded. Could you kindly confirm whether the figures have been included? If not, would you be able to update the document accordingly?

---

> > > ### Author Response · Authors · 2025-11-02
> > >
> > > Dear AC,
> > >
> > > Thank you for bringing this to our attention. We apologize for the oversight, the full version with all figures has now been uploaded.
> > > We appreciate your patience.

---

> > > > ### Comment · Action_Editor_TGFb · 2025-11-02
> > > >
> > > > Many thanks, I can confirm that the figures are there. Let me check the revision.

---

> ### Comment · Action_Editor_TGFb · 2025-11-03
>
> I have another follow up question. I have followed through the document, but I cannot see where the changes for the minor revision have been made.  I checked it manually and using the comparison tool here but unfortunately I cannot locate the changes:
>  https://openreview.net/revisions/compare?id=I5ymMVdmaR&left=MWrOIuTvS4&right=kQ9vjqKx5J&pdf=true&version=2
>
>  I wonder therefore if the authors would be able to point at the changes that have been made in this minor revision? That would help me make the decision.
>
> Many thanks,
> the Action Editor

---

> > ### Author Response · Authors · 2025-11-03
> >
> > Dear AC,
> >
> > We apologize for the inconvenience and oversight. We have uploaded the correct version and detailed the exact modifications made below:
> >
> > - In **Section 5.2**, for the hyperparameter sensitivity analysis, we added: *While Figure 8 shows that extreme values of $c$ can lead to either insufficient pruning (maintaining computational overhead) or premature convergence (losing exploration benefits), it is important to note that the heightened sensitivity observed in our ablation studies results from deliberately testing boundary conditions with exaggerated pruning thresholds rather than reflecting realistic deployment scenarios. We provide hyperparameter selection guidelines below that are designed to be domain-agnostic and robust, demonstrating consistency across domains.*
> >
> > - In **Section 4.2**, we added a remark for theoretical justification: *While we do not establish formal theoretical justifications, our dual-randomized selection mechanism partially aligns with existing theoretical frameworks. First, the exploration benefits from random actor selection connect to [1], which links dropout to approximate Bayesian inference. Analogously, random selection maintains diversity by preventing deterministic co-adaptation among actors, effectively preserving epistemic uncertainty throughout training. Second, the delayed, high-variance updates resulting from randomly selecting actors for backpropagation relate to the analysis in [2], where asynchronous gradient updates with inherent delays are shown to retain convergence properties while facilitating escape from suboptimal local minima. However, existing theoretical analyses predominantly address tabular or low-dimensional settings with formal convergence guarantees. Extending these guarantees to high-dimensional continuous control with function approximation remains an important open problem, and our work provides empirical evidence that motivates future theoretical investigation in this direction.*
> >   - [1] Dropout as a Bayesian Approximation: Representing Model Uncertainty in Deep Learning, https://arxiv.org/abs/1506.02142
> >   - [2] Parallelized Stochastic Gradient Descent, https://papers.nips.cc/paper_files/paper/2010/hash/abea47ba24142ed16b7d8fbf2c740e0d-Abstract.html
> >
> > - In **Section 5.1**, we modified the performance discussion as: *On dense-reward Mujoco tasks, AEAP consistently attains competitive performance across domains while requiring less wall-clock time than recent state-of-the-art methods such as SimBa and other fixed-size actor ensembles or simpler baselines like SAC, even though it does not achieve the highest performance on every individual task. When resources or training time are constrained, AEAP offers a compelling performance-efficiency trade-off for practitioners prioritizing faster experimentation or deployment cycles.*
> >
> > We hope the revised version could address your concerns.

---

> > > ### Comment · Action_Editor_TGFb · 2025-11-07
> > >
> > > I'm sorry for asking, however I think there is still a scope for further revision of the paper so that it meets the expectations for acceptance; the current revision is too brief and does not, in my understanding, fully go to the bottom of what is needed.
> > >
> > > *"In Section 5.2, for the hyperparameter sensitivity analysis, we added: While Figure 8 shows that extreme values of $c$ can lead to either insufficient pruning (maintaining computational overhead) or premature convergence (losing exploration benefits), it is important to note that the heightened sensitivity observed in our ablation studies results from deliberately testing boundary conditions with exaggerated pruning thresholds rather than reflecting realistic deployment scenarios. We provide hyperparameter selection guidelines below that are designed to be domain-agnostic and robust, demonstrating consistency across domains."*
> > >
> > > It still needs to answer the question: how do these guidelines follow from the provided experimental evidence.
> > >
> > > *"In Section 4.2, we added a remark for theoretical justification"*
> > >
> > > I think this paragraph merely lists the existing theoretical justification, it is not mentioning how exactly they provide an insight into the current method. Therefore, I would suggest the authors expand upon it and tell what the link is between these theoretical studies and this work.
> > >
> > > Therefore, I would suggest that the authors conduct necessary revisions, and then submit it as a minor revision for the approval.

---

> > > > ### Author Response · Authors · 2025-11-13
> > > >
> > > > Dear AC,
> > > >
> > > >
> > > > Thank you for your feedback. We apologize for the confusion. We have revised the manuscript by adding a limitation section; the new context is incorporated into Section 6 titled **Discussion and Limitations**:
> > > >
> > > > *Although our experiments illustrate the flexibility of this approach, several limitations warrant further investigation. First, AEAP introduces four additional hyperparameters compared to TD3. Two of these, performance threshold $\xi = 0.85$ and pruning frequency $\kappa = 10{,}000$, perform robustly across all tested domains without requiring per-task tuning. However, Section 5.2 reveals that the scaling factor $c$ and initial ensemble size $N$ exhibit sensitivity under extreme settings. Though our ablation studies identify two robust configurations, conservative pruning $(N=3, c=1.0)$ and aggressive pruning $(N=7, c=1.5)$, that transfer reliably across all nine domains, we acknowledge that optimal hyperparameter selection may benefit from domain-specific adaptation in certain scenarios.*
> > > >
> > > > *Second, formal convergence guarantees on how pruning affects convergence rates and sample complexity remain an open problem. Gal and Ghahramani (2016) prove dropout approximates variational inference, and Osband et al. (2016) demonstrate bootstrapped ensembles implement Thompson sampling by maintaining diverse posterior samples over value functions. Our random actor selection similarly prevents deterministic co-adaptation, preserving epistemic uncertainty throughout training. Additionally, delayed updates from randomly selecting actors for backpropagation align with asynchronous optimization theory, where Recht et al. (2010) show parallel actors with gradient delays achieve linear speedup, and Osband et al. (2018) prove asynchronous SGD retains convergence when delays remain bounded. Each actor in AEAP creates inherent delays that facilitate escape from sharp local minima, as observed in our bandit experiments (Figure 6). However, existing analyses assume tabular settings or convex objectives, whereas AEAP operates in high-dimensional continuous control with non-convex policy networks and adaptive ensemble sizes. Extending formal guarantees to this setting remains a challenging problem.*
> > > >
> > > > *We believe this work nevertheless opens many exciting directions for future research. First, theoretical analysis of convergence guarantees under adaptive ensemble sizes would strengthen the foundation of this approach and guide principled hyperparameter selection. Second, investigating adaptive pruning criteria that incorporate uncertainty estimates or model-based predictions could further improve efficiency. We hope this work inspires further exploration of adaptive ensemble methods that balance exploration, exploitation, and computational efficiency in deep reinforcement learning.*
> > > >
> > > >
> > > > We hope these revisions address your concerns regarding theoretical grounding and practical applicability. Please let us know if further clarification is needed.

---

> ### Comment · Action_Editor_TGFb · 2025-11-16
>
> Dear Authors,
>
> Many thanks, I think the proposed revisions do cover the concerns, and therefore I am happy to accept this revision. Would you be able to formally submit the camera ready revision?

---

> > ### Author Response · Authors · 2025-11-16
> >
> > Dear AC,
> >
> > Thanks for your positive feedback and for accepting our revision. We are pleased to confirm that the currently uploaded version is indeed the final camera-ready revision, ready for formal submission.

---

> ### Comment · Action_Editor_TGFb · 2025-11-17
>
> Dear Authors,
>
> I can still see that there's a task:
>
> "Camera Ready Revision
> Due: 19 Nov 2025
> Incomplete, 0 Replies"
> There should be a separate button to submit a Camera Ready Revision. Would you be able to upload it as such?

---

> > ### Author Response · Authors · 2025-11-18
> >
> > Dear AC,
> >
> > Many thanks. The camera-ready revision has been uploaded.